# GLOV: Guided Large Language Models as Implicit Optimizers for Vision Language Models

## Abstract

In this work, we propose a novel method (GLOV) enabling Large Language Models (LLMs) to act as implicit Optimizers for Vision-Language Models (VLMs) to enhance downstream vision tasks. Our GLOV meta-prompts an LLM with the downstream task description, querying it for suitable VLM prompts (*e.g.,* for zero-shot classification with CLIP). These prompts are ranked according to their fitness for the downstream vision task. In each respective optimization step, the ranked prompts are fed as in-context examples (with their accuracies) to equip the LLM with the knowledge of the type of prompts preferred by the downstream VLM. Furthermore, we also explicitly steer the LLM generation in each optimization step by specifically adding an offset difference vector of the embeddings from the *positive* and *negative* solutions found by the LLM, in previous optimization steps, to the intermediate layer of the network for the next generation step. This offset vector steers the LLM generation toward the type of language preferred by the downstream VLM, resulting in enhanced performance on the downstream vision tasks. We comprehensively evaluate our GLOV on 16 diverse datasets using two families of VLMs, *i.e.,* dual-encoder (*e.g.,* CLIP) and encoder-decoder (*e.g.,* LLaVa) models – showing that the discovered solutions can enhance the recognition performance by up to 15.0% and 57.5% (3.8% and 21.6% on average) for these models.

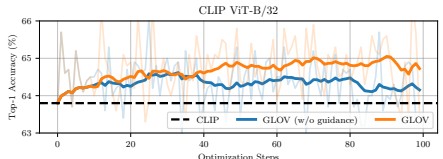 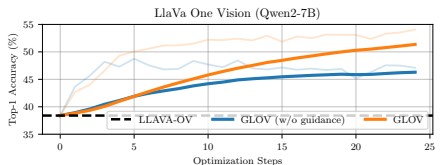

Figure 1: **The effect of prompt evolution on the downstream task performance.** The shaded regions represent the absolute top-1 classification accuracies for ImageNet (Deng et al., 2009) at each optimization step by ensembling the top-3 prompts found w.r.t the accuracy on the 1-shot train set whereas the solid lines represent the exponential moving average. The left plot is with CLIP VIT-B/32 (Radford et al., 2021), and the right is with LLaVa-OV (Li et al., 2024) while the LLM employed is Llama-3 (Dubey et al., 2024). Due to high computational cost, we only perform 25 optimization steps for LLaVa-OV.

## 1 Introduction

Orthogonal to traditional gradient-based optimization (Nesterov, 1983; Boyd & Vandenberghe, 2004; Kingma & Ba, 2014; Ruder, 2016), the recent rise of large language models (Brown et al., 2020; OpenAI, 2023; Chiang et al., 2023; Raffel et al., 2020; Touvron et al., 2023a;b; Dubey et al., 2024) and vision-language foundation models (OpenAI, 2023; Li et al., 2024; Zhu et al., 2024; Alayrac et al., 2022; Radford et al., 2021) has introduced the possibility of framing optimization in the context of natural language prompts. This form of optimization typically does not require any gradient-based learning or parameter update but focuses on extracting knowledge from the language models via suitable natural language prompts. A large body of work focuses on finding natural language prompts optimized for various downstream tasks for both LLMs (Yang et al., 2024; Wei et al., 2022; Kojima et al., 2022; Yao et al., 2023) and VLMs (Pratt et al., 2023; Roth et al., 2023; Mirza et al., 2024), demonstrating impressive gains in language-based and downstream vision tasks.

In our work, we frame optimization around discovering suitable natural language prompts for VLMs, with the objective of improving performance on downstream vision tasks. Our proposed GLOV employs a prompt search technique relying on a meta-prompt coupled with embedding space guidance, that drives the

prompt optimization for the VLMs. We use the meta-prompt to iteratively query an LLM with downstream task-specific description and ranked in-context examples derived from the previous (optimized) prompts. In-context examples guide the LLM toward the desired output, and their ranking (measured on a small held-out train set) provides the LLM with a sense of the language patterns preferred by the downstream VLM. To further steer the LLM generation towards a notion of *goodness* in each optimization step, we explicitly bias the language generation with a direction. The direction is determined by adding a hidden state offset vector (on the last token during the autoregressive generation) derived from the *positive* and *negative* prompts (based on their effectiveness on labeled training data) to the LLM's activation space during generation. The intuition is that by directing the LLM generation toward the *positive* prompts, the model can discover semantically similar and potentially more effective solutions. One complete optimization run, depicting the effectiveness of the discovered solutions and the effect of applying the embedding space guidance is plotted in Figure 1. The best-performing prompts (on the held-out train set) achieve an absolute improvement of 2.6% and 15.2% on ImageNet (Deng et al., 2009) test set over CLIP (Radford et al., 2021) and LLaVa-OV (Li et al., 2024) respectively.

We extensively evaluate our GLOV on one of the fundamental tasks in computer vision: image classification, and also touch upon the open-ended generation task of visual question answering (VQA). We demonstrate the generalization of our GLOV on a total of 16 diverse datasets, with the two commonly employed families of VLM models – the dual-encoder and the recent visual encoder-decoder models (Radford et al., 2021; Li et al., 2024). We find that our GLOV can consistently discover highly effective solutions for the downstream task of interest resulting in significant improvements across the board. For example, the most effective prompts discovered for the dual-encoder models (*e.g.,* CLIP) can improve the accuracy up to 15.0% (3.81% on average) and for the encoder-decoder architectures (*e.g.,* LLaVa), the resulting prompts show an even larger improvement of up to 57.5% (21.6% on average). Furthermore, we extensively ablate our proposed prompt optimization algorithm, design choices, and the effect of our guidance mechanism, providing insights for future work.

## 2 RELATED WORK

Our work is related to large language and vision language models, approaches proposing methods for steering the LLM outputs, and prompt optimization methods (through LLMs) for VLMs.

### 2.1 LLMS AND VLMS

Here, we first provide a brief overview of Large-Language Models (LLMs) and then move towards Vision-Language Models (VLMs).

**LLMs** have revolutionized the natural language processing landscape. These models can typically be divided into two major groups; namely the long-short-term-memory (LSTM) (Hochreiter & Schmidhuber, 1997) and transformer-based architectures (Vaswani et al., 2017). The former is based upon recurrent neural networks (RNNs) that use gates to control the flow of information, allowing them to capture long-term dependencies in sequential data. The latter is based on the self-attention mechanism, which enables the model to process sequences in parallel and capture relationships between tokens regardless of distance. Some notable works following the LSTM family of models include (Sutskever et al., 2014; Graves & Schmidhuber, 2005; Bahdanau et al., 2015; Beck et al., 2024). The transformer-based architectures consist of encoder or decoder-based LLMs. The encoder-based LLMs are primarily used for understanding tasks like text classification and sentiment analysis, as they excel at capturing contextual information from the input. Some notable works include BERT (Devlin et al., 2019), RoBERTa (Liu et al., 2019b), and DistilBERT (Sanh et al., 2020). The decoder-based LLMs, on the other hand, are designed for generative tasks such as text generation, translation, and summarization, with recent models including GPT-3 (Brown et al., 2020), T5 (Raffel et al., 2020), GPT-4 (OpenAI, 2023), and the Llama family of models (Dubey et al., 2024). In our work, since we need to access the weights of the LLMs, we resort to the open-source Llama-3 model. However, potentially any open-source LLM can be employed.

**VLMs** can be placed in two categories. One group relies on dual-encoders (vision and text encoder), usually trained in a contrastive manner and these models are typically strong at tasks like image recognition. The most common among these methods are CLIP (Radford et al., 2021), ALIGN (Jia et al., 2021), OpenCLIP (Schuhmann et al., 2022), SigLIP (Zhai et al., 2023), and MetaCLIP (Xu et al., 2023). Many methods (Mirza et al., 2024; 2023b; Doveh et al., 2023b;a; Lin et al., 2023; Mirza et al., 2023a) build upon these models to further improve them for specific downstream tasks. The other group of methods

aligns the visual modality with a frozen LLM and can be used for open-ended visual reasoning tasks like image captioning, visual question-answering, etc. Some representative approaches from this group include BLIP-2 (Li et al., 2023), Instruct-BLIP (Dai et al., 2023), MiniGPT (Zhu et al., 2024; Chen et al., 2024), and the LLaVa family of models (Liu et al., 2023; Li et al., 2024). Similarly, some approaches (Doveh et al., 2024; Gavrikov et al., 2024; Lin et al., 2024; Huang et al., 2024) build upon these models and provide further improvements. In our work, we focus on the task of object recognition by employing both families of models and frame the task of finding the optimal prompt templates (for CLIP (Radford et al., 2021)) and suitable prompts for open-ended generation (for LLaVa (Li et al., 2024)) as an optimization problem. Specifically, for the decoder-based VLMs, some recent works (Zhang et al., 2024) highlight that these models struggle for fine-grained object recognition. However, we show in our work, for the first time, that our GLOV can discover an optimal prompt that can greatly improve the visual recognition ability of these models, without requiring any gradient-based learning or fine-tuning.

## 2.2 STEERING LLM RESPONSES

One line of work alters the responses from an LLM, without requiring any explicit gradient-based fine-tuning. ActADD (Turner et al., 2023) proposes to reduce LLM hallucinations by altering the hidden states. In their work, given a positive and a negative data point (model response), they propose to steer the responses (*e.g.,* to be less *hateful*) by adding the difference of the embeddings from these points to the intermediate layers of the network. On the other hand, Proxy-Tuning (Liu et al., 2024a) proposes to adapt the model responses on the logit level. Specifically, they model the responses from a smaller base model to be similar to a larger instruction-tuned LLM by altering the softmax probabilities obtained from the smaller base model. We also take inspiration from Turner et al. (2023); Liu et al. (2024a) and for the first time show that such steering (applied on the embedding level) can also be used to improve downstream multi-modal (vision-language) tasks. One difference between ActADD and our GLOV is that we add the difference of the sentence embeddings (and not of the prompts themselves, as in ActADD) to only the last token. Whereas, ActADD adds the difference of the (complete) sequence lengths to the first few tokens at the generation step. In the ablations Section 4.3, we show that this seeming nuance has an important effect on the performance of the downstream task, showing that our method might be more suitable for vision tasks.

## 2.3 LARGE-LANGUAGE MODELS AS PROMPT OPTIMIZERS

Some approaches propose employing LLMs (in an agentic workflow) to search for the optimal prompt for the downstream task. OPRO (Yang et al., 2024) coins the term "LLMs as Optimizers" and proposes iteratively discovering solutions (prompts) for natural language tasks by employing an LLM in a feedback loop. Similarly, Liu et al. (2024b) proposes to find suitable prompts for dual-encoder VLMs (*e.g.,* CLIP) by iteratively prompting an LLM. Our GLOV also proposes to discover suitable prompts for VLMs but differs from Liu et al. (2024b) in the sense that we are employing a meta-prompt that captures long-range dependencies by tapping into the history-buffer of the in-context examples and employs task-specific knowledge that helps to obtain prompts better suited for the downstream task. Furthermore, we propose a new method for steering the LLM generation (through embedding space guidance) towards the responses that are more suitable for downstream VLMs. Powered by a suitable meta-prompt and the guidance scheme, our GLOV discovers solutions which help to enhance visual tasks for both the dual-encoder and encoder-decoder models.

## 3 GLOV: GUIDED LLMs AS IMPLICIT OPTIMIZERS FOR VLMs

The goal of our GLOV is to improve the VLM's downstream (vision) task performance by optimizing natural language prompts through employing an LLM in an iterative workflow. To achieve this, we build upon a meta-prompt introduced by Mirza et al. (2024), differing from them, we leverage few-shot (*e.g.,* 1-shot) held-out labeled training examples to calculate the effectiveness of the solutions discovered in each optimization step, which guides the optimization. Furthermore, effective prompt optimization is performed by providing the LLM with explicit guidance conditioned on a prior of the difference of the sentence embeddings from the *positive* and *negative* prompts discovered during the previous optimization iterations. Although the application space of GLOV is general and we demonstrate its generalization ability on two popular families of VLMs (*e.g.,* dual-encoder (Radford et al., 2021) and encoder-decoder (Liu et al., 2023)), for simplicity, here we focus our description around CLIP (Radford et al., 2021) while mentioning the differences for LLaVa (Li et al., 2024) where appropriate. An overview of our methodology is provided in Figure 2.

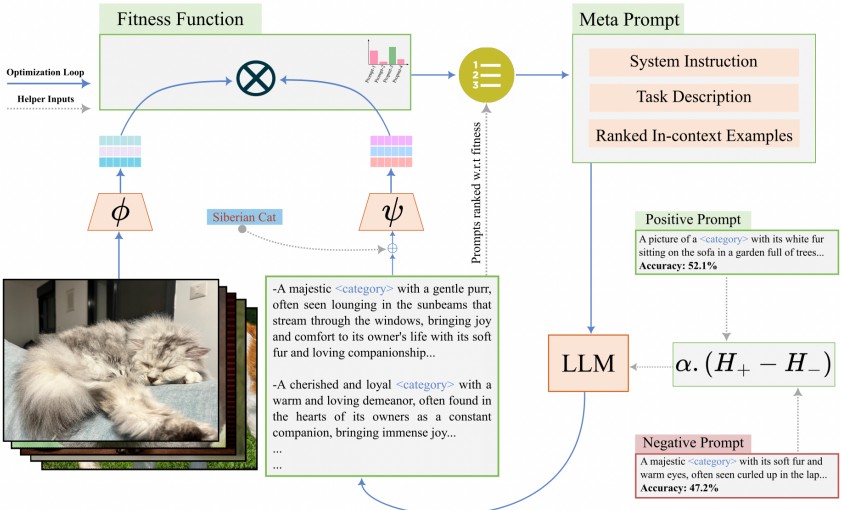

Figure 2: **Overview of GLOV**. GLOV consists of a Meta Prompt, which constitutes system instruction, task description, and in-context examples (VLM prompts) which are evaluated (and ranked) on a few-shot training data in each iteration. The Meta-Prompt instructs the LLM to generate several candidate solutions in each optimization iteration, conditioned on the in-context examples which are fed in conjunction with the accuracy values, highlighting their effectiveness. Furthermore, to steer the LLM generation towards the language preferred by the VLM, we add the scaled difference of the sentence embeddings (autoregressively) from the *positive* and *negative* text prompts to the intermediate layer of the LLM. This process is repeated until the stopping condition is met (*e.g.,* maximum iterations). Note, that $H_+$ and $H_-$ refer to the sentence embeddings from the text prompts.

For the ease of assimilation, we divide the description of our GLOV into different parts. In Section 3.1 we describe the fitness function and how it can provide an interface for the LLM-VLM interaction. In Section 3.2, we provide details about the meta-prompt employed in our work. Finally, we conclude in Section 3.3 by providing details about the proposed guidance methodology.

## 3.1 LLM-VLM INTERACTION THROUGH FITNESS FUNCTION

The dataset-specific prompt templates $\mathcal{P}$ provided by CLIP (Radford et al., 2021) have been constructed manually, requiring human effort. In this work, we frame the prompt search as an optimization problem and propose to replace the human with an LLM, employed in an iterative feedback loop. Furthermore, we explicitly guide the generation process of the LLM in each optimization step by proposing a novel guidance methodology that can assist the LLM in understanding the style of language preferred by the downstream VLM, even though the two models only interact through a fitness function. At each optimization step $i$, the LLM provides multiple (*e.g.,* 10) solutions to improve the downstream task performance. However, not all solutions provided by the LLM are preferred for the downstream vision task. To obtain a measure of the fitness (effectiveness) of the provided solutions to the downstream vision task, we evaluate all the candidate solutions on a held-out few-shot (1-shot) labeled training dataset $\mathcal{D}$. For CLIP (Radford et al., 2021), the zero-shot likelihood of class $\hat{c}$ for each discovered prompt $p \in \mathcal{P}$ during an optimization step can be found by

$$l_{\hat{c}}(x) = \frac{e^{\cos(\psi_{\hat{c}}, \phi(x))/\tau}}{\sum_{c \in C} e^{\cos(\psi_c, \phi(x))/\tau}}, \quad \text{where} \quad \psi_c = \psi(p(c)), \tag{1}$$

where $\phi$ and $\psi$ represent the vision and text encoders of CLIP, $x \in \mathcal{D}$, $\tau$ denotes the temperature constant, cos refers to the cosine similarity, and $p(c)$ replaces the 'class' placeholder in the discovered prompt $p$. CLIP's vision encoder $\phi$ produces the image embedding. The text embedding of a class $c$ (belonging to a set of candidate classes $C$) is obtained by incorporating the class name $c$ in the found prompt, called a VLM prompt, and embedding this text through the VLM's text encoder $\psi$. The fitness of a prompt $p \in \mathcal{P}$ can be found by comparing the predicted label with the ground truth, summarized as

$$\text{Fitness}(p) = \frac{1}{|\mathcal{D}|} \sum_{(x,y)\in\mathcal{D}} \mathbb{1}\left[\underset{c}{\operatorname{argmax}} \, l_c(x) = y\right], \tag{2}$$

where $\mathbb{1}$ is an indicator function that is 1 if the predicted label matches the ground truth $y$ and 0 otherwise. For the encoder-decoder models (*e.g.,* LLaVa), which produce open-ended outputs, we obtain the class likelihoods by obtaining a symbolic representation of the image in textual form and comparing the text embeddings from this symbolic representation with the text embeddings obtained for the individual class names, using a dedicated sentence embedding model (Reimers & Gurevych, 2019). We expand on these details in the Appendix Section A.

It is important to note that the fitness function forms a bridge between two disjoint models – the LLM and the VLM, and is responsible for their interaction. The fitness (classification accuracy) provides feedback to the LLM regarding the type of natural language sentences that are preferred by the downstream VLM. The fitness function is responsible for ranking all the prompts provided as in-context examples to the meta-prompt in each optimization iteration (*c.f.,* Section 3.2) and also forms the basis for the application of the embedding-space guidance methodology (*c.f.,* Section 3.3) proposed in this work to bias the LLM responses towards a notion of *goodness*.

### 3.2 META-PROMPT

The meta-prompt (*c.f.,* Appendix Figure 5) is responsible for driving the iterative prompt optimization. Specifically, it consists of 3 distinct parts, which are described as follows:

**System prompt** is a generic set of instructions that describe the task of the LLM. It helps a user to find the optimal prompts, improving the downstream task accuracy. The system prompt remains static for the entire optimization run.

**Task description** is dynamically changing at each optimization step. It consists of a description of what is included in the main body of the prompt, *e.g.,* what is expected from the LLM (*i.e.,* a prompt), a downstream task name, task description, quantity (and actual) best and worst prompt templates as in-context examples, with their associated fitness, obtained through equation 2.

**In-context examples** serve to bootstrap the LLM to the type of output that is expected. In each optimization step, we task the LLM to provide us with 10 candidate solutions (prompts). Each prompt is evaluated w.r.t. a fitness function to obtain a (classification) score. We keep a global history of the prompts (and associated fitness) generated during all the previous optimization steps and at each optimization step $i$, the newly generated prompts and all the previous prompts are ranked according to their respective fitness score. For the next optimization step $i+1$, $top_k$ and $bottom_k$ prompts (we choose $k=5$) are selected as in-context demonstrations and plugged into the meta-prompt together with their respective accuracies. The intuition behind keeping a global history of prompts is to provide the LLMs with long-term knowledge about what types of prompts have been effective for the VLM so that it can model its responses according to them. The motivation behind the current choice of the $top_k$ and $bottom_k$ in-context examples is that we intend to provide contrasting examples to the LLM from the opposite end of the spectrum (of *goodness* and *badness*) so that the LLM can make sense of what are the type of responses preferred by the LLM.

### 3.3 STEERING THE LLM GENERATION PROCESS

At a higher level, given two prompts – *positive* and *negative* (identified through equation 2), our proposed steering can be considered as analogous to computing a *hidden state gradient* towards the *positive* prompt, effectively biasing the language generation away from the *negative* identified prompt in each optimization step. The intuition is to condition the LLM text outputs according to the language preferred by the downstream VLM. To this end, we show that the LLM outputs can be steered through simple arithmetic in the hidden states of the present-day LLMs.

For a given LLM $f$ with pre-trained parameters $\vec{\theta}$, and given tokenized prompts $\vec{p_b}$ and $\vec{p_w}$, the activation responses at layer $l$ are denoted as $a_l(\vec{p};\vec{\theta})$. This is an activation map of $B \times S \times E$, which denotes the number of prompts $B$, tokenized sequence length $S$, and the hidden dimension size $E$. Typically, $a_l$ does not depend on all model parameters $\vec{\theta}$, but we abuse the notation in the interest of simplicity. The sentence embeddings $H_+$ and $H_-$ can be obtained by averaging the activations across the sequence length $S$

| | ImageNet | ImageNetv2 | Caltech101 | ImageNetR | ImageNetS | ImageNetA | OxfordFlowers | OxfordPets | Mean |
|---|---|---|---|---|---|---|---|---|---|
| CLIP (S-TEMP) (Radford et al., 2021) | 61.9 | 54.8 | 91.4 | 65.4 | 40.3 | 28.2 | 64.0 | 81.3 | - |
| CLIP (DS-TEMP) (Radford et al., 2021) | 63.3 | 56.0 | 89.9 | 67.9 | 42.1 | 30.2 | 66.6 | 83.2 | - |
| LLM-OPT (Liu et al., 2024b) | 62.8 | 55.6 | 92.3 | 67.5 | 41.9 | 28.1 | 67.0 | 78.1 | - |
| GLOV (w/o guidance) | 62.7 | 55.8 | 92.1 | 67.8 | 41.9 | 31.2 | 64.6 | 84.4 | - |
| GLOV | 64.5 | 56.6 | 93.7 | 68.5 | 43.0 | 32.5 | 67.7 | 85.5 | - |
| | StanfordCars | DescribableTextures | Food101 | FGVCAircraft | SUN397 | UCF101 | RESISC45 | EuroSAT | |
| CLIP (S-TEMP) (Radford et al., 2021) | 60.2 | 40.2 | 77.6 | 18.1 | 62.1 | 60.4 | 54.1 | 35.8 | 55.8 |
| CLIP (DS-TEMP) (Radford et al., 2021) | 59.9 | 42.4 | 79.2 | 19.4 | 61.7 | 62.3 | 57.2 | 45.8 | 57.9 |
| LLM-OPT (Liu et al., 2024b) | 60.2 | 41.7 | 79.2 | 17.7 | 60.9 | 60.9 | 54.4 | 45.0 | 57.1 |
| GLOV (w/o guidance) | 59.6 | 41.4 | 78.5 | 19.7 | 62.2 | 63.0 | 61.4 | 46.9 | 58.3 |
| GLOV | 60.4 | 42.6 | 79.5 | 20.1 | 62.1 | 63.8 | 62.0 | 50.8 | 59.6 |

Table 1: **Results on dual-encoder VLM.** Top-1 accuracy (%) for 16 datasets obtained by employing the ViT-B/32 backbone from OpenAI CLIP (Radford et al., 2021). *S-TEMP* refer to the results obtained by using the default template (a photo of a <class name>), while *DS-TEMP* refer to the results obtained by using the ensemble of dataset-specific prompts. GLOV (w/o guidance) represents the results without the *guidance* applied to the LLM generation, whereas GLOV represents results obtained by adding the guidance offset vector. The mean results over 20 datasets are reported in the bottom half of the table. The **bold** numbers represent the best and the underline numbers represent the second-best accuracy.

$$H_+ = \frac{1}{S_+}\sum_{s=1}^{S_+} a_l(\vec{p_+};\vec{\theta})_{:,s,:} \quad H_- = \frac{1}{S_-}\sum_{s=1}^{S_-} a_l(\vec{p_-};\vec{\theta})_{:,s,:} \tag{3}$$

where $S_+$ and $S_-$ are the sequence lengths of prompts $\vec{p_+}$ and $\vec{p_-}$, respectively. The goal is to obtain semantically meaningful sentence embeddings from the identified *positive* and *negative* prompts.

For each new token produced in the subsequent optimization iteration, the difference between $H_+$ and $H_-$ is added autoregressively to the embeddings of each generated token[1]. Let $\vec{p}_n$ denote the new token appended to the (meta) prompt, then the updated sentence embedding $H_n$ is given by

$$H_n = H_n + \alpha \cdot (H_+ - H_-) \tag{4}$$

where $\alpha$ is the scaling factor and is chosen via grid search. This process is repeated until the maximum number of tokens is achieved for each prompt. In total we prompt the LLM (at each iteration) to provide us with 10 prompt templates for CLIP and 5 for LLaVa to reduce the computation efforts. In an optimization run $p_+$ is always the prompt with the best accuracy w.r.t the fitness and $p_-$ is set to be the prompt with the second-best accuracy. Since, we compute a form of the gradient-like differential between averages of token hidden states, intuitively trying to identify a characteristic of task-specific improvement. Thus, the intuition behind computing the differential between the best and the second best (in terms of fitness) is to make it between points closest to the maximal value of the objective – which is a common mathematical intuition. Furthermore, $p_+$ and $p_-$ are only updated when a new prompt with higher accuracy is found. This ensures that the guidance signal does not alter in each iteration, resulting in more stable optimization.

An important design choice in GLOV is the method adopted to calculate the sentence embeddings. Some works, *e.g.,* Jiang et al. (2023) hint that the decoder-based LLMs are not suitable for obtaining the sentence embeddings. We ablate our proposed method of obtaining sentence embeddings (equation 3) in Section 4.3 and find it to provide strong results (while linear probing the embeddings from the middle layers of the LLM) on common natural language classification tasks, hinting that our sentence embeddings can capture semantically meaningful information from the prompts.

## 4 EXPERIMENTAL EVALUATIONS

In this section, we first provide a brief overview of the datasets we use for evaluating our GLOV, then provide an overview of the different baselines and state-of-the-art methods we compare to, later discuss the implementation details and finally conclude with a discussion on the results.

### 4.1 EVALUATION SETTINGS

**Datasets:** We extensively evaluate our GLOV on 16 object recognition datasets belonging to widely different domains. These domains can be narrowed down to datasets containing commonly occurring

---

[1] We also experiment with several alternatives for adding the offset vector in the ablations Section 4.3, however, we find that adding the offset to *only* the last token performs best.

| | ImageNet | ImageNetv2 | Caltech101 | ImageNetR | ImageNetS | ImageNetA | OxfordFlowers | OxfordPets | Mean |
|---|---|---|---|---|---|---|---|---|---|
| LLaVA-OV (Li et al., 2024) | 36.5 | 31.4 | 77.7 | 52.1 | 38.1 | 32.3 | 19.4 | 16.2 | - |
| Meta-Prompt (Mirza et al., 2024) | 45.0 | 42.5 | 86.1 | 64.9 | 45.9 | 42.9 | 28.5 | 53.7 | - |
| GLOV (w/o guidance) | 46.8 | 40.9 | 87.1 | 75.7 | 49.6 | **44.8** | 28.6 | 53.7 | - |
| GLOV | **51.7** | **46.1** | **92.6** | **77.6** | **49.9** | 43.6 | **39.6** | **54.3** | - |

| | StanfordCars | DescribableTextures | Food101 | FGVCAircraft | SUN397 | UCF101 | RESISC45 | EuroSAT | - |
|---|---|---|---|---|---|---|---|---|---|
| LLaVA-OV (Li et al., 2024) | 21.7 | 33.2 | 21.5 | 4.1 | 36.4 | 52.9 | 43.3 | 25.6 | 33.9 |
| Meta-Prompt (Mirza et al., 2024) | 65.4 | 49.0 | 58.1 | 39.7 | 40.0 | 55.1 | 49.4 | **41.4** | 50.4 |
| GLOV (w/o guidance) | 73.9 | 46.9 | 66.9 | **44.0** | 44.9 | **60.6** | 47.2 | 36.3 | 52.9 |
| GLOV | **79.2** | **51.7** | **67.0** | 41.0 | **46.0** | 59.7 | **51.1** | 36.3 | **55.5** |

Table 2: **Results on encoder-decoder VLM.** Top-1 accuracy (%) for 16 datasets obtained by employing the LLaVa (One Vision) (Li et al., 2024). *LLaVa (OV)* refer to the results obtained by using a generic prompt, while *Meta-Prompt* refer to the results obtained by obtaining the prompts through Mirza et al. (2024). The mean results over 20 datasets are reported in the bottom half.

**natural categories**: ImageNet (Deng et al., 2009), ImageNetV2 (Recht et al., 2019), Caltech101 (Fei-Fei et al., 2004), **fine-grained** classification datasets containing different task-specific images: Oxford Flowers (Nilsback & Zisserman, 2008), Standford Cars (Krause et al., 2013), Oxford Pets (Parkhi et al., 2012), Describable Textures Dataset (DTD) (Cimpoi et al., 2014), Food-101 (Bossard et al., 2014), FGVC-Aircraft (Maji et al., 2013). Dataset used for **scene classification**: SUN397 (Xiao et al., 2010), **action recognition dataset**: UCF101 (Soomro et al., 2012). Datasets consisting of **out-of-distribution images**: ImageNet-(R)endition (Hendrycks et al., 2021a), ImageNet-(A)dversarial (Hendrycks et al., 2021b), ImageNet-(S)ketch (Wang et al., 2019) and also datasets which contain images taken from a **satellite or an aerial view**: EuroSAT (Helber et al., 2018) and RESISC45 (Cheng et al., 2017).

**Baselines:** We compare to the following baselines and state-of-the-art methods:

**CLIP** (Radford et al., 2021) denotes the zero-shot classification scores obtained by using the simple '`{a photo of a <class name>}`' template (S-TEMP) and dataset-specific templates (DS-TEMP[2]). **LLaVa-OV** represent results obtained by using a base prompt[3]. **LLM-OPT** (Liu et al., 2024b) proposes to find suitable text prompts for the downstream datasets by iterative refinement through an LLM. However, their method relies only on the in-context examples without explicit guidance[4]. **Meta-prompt** (Mirza et al., 2024) propose a method for improving the visual recognition performance of dual-encoder models by generating category-level VLM prompts. Here we extend its evaluations to dual-encoder models[5]. For completeness, we also compare with the gradient-based (few-shot) learning method **CoOp** (Zhou et al., 2022) in the Appendix Table 5.

**Implementation Details:** To report the results for each dataset we use the test splits provided by Zhou et al. (2022). All the baselines are also implemented in the same framework. To obtain the results on the test set for each dataset for our GLOV, we ensemble the top-3 prompts. These prompts are chosen with regard to the best-performing prompts on the 1-shot train set at a certain iteration during the optimization. For our GLOV we use Llama-3 (Touvron et al., 2023a) from Hugging Face, whereas, for LLM-OPT we keep consistent with their original implementation regarding all details. The best-performing prompts for LLM-OPT are also chosen w.r.t the 1-shot train set. We set the maximum number of optimization iterations to 100 (with 10 candidate solutions at each iteration) for the experiments with CLIP and 50 (with 5 candidate solutions) for LLaVA-OV, except for datasets containing 1000 classes (*e.g.,* ImageNet), where we set the maximum number of iterations to 25 to save computation time. In general, the experiments with CLIP can run on a single NVIDIA 3090 (24GBs) GPU, and the experiments with LLaVa fit on an A40 (48GBs) GPU or similar. Our entire codebase is attached as `.zip` file with the supplementary material.

---

[2]https://github.com/openai/CLIP/blob/main/data/prompts.md

[3]The prompt used to obtain the base results is: `Describe the category present in this image briefly and also identify the name of the category present,` which was able to match or surpass accuracy reported by Zhang et al. (2024).

[4]Note that this method is not suitable for application on the encoder-decoder models because it requires a memory bank of templates, which are not available *a-priori* for these models.

[5]The original publication generates category-level prompts for the dual-encoder models, which is a different setting than the one studied in our work.

## 4.2 RESULTS

We evaluate our GLOV extensively on 16 diverse datasets. In Table 1 we list the results by employing the CLIP ViT-B/32 from OpenAI. We observe that our GLOV achieves better accuracy on all the datasets evaluated. For example, as compared to CLIP, when using the simple prompt template, our vanilla GLOV provides an average gain of 2.5%, with up to 11.0% gains on EuroSAT. Similarly, the gains increase even further with our proposed guidance scheme. We observe gains of up to 15% (3.8% on average). On the large-scale ImageNet dataset, we observe gains of 2.5%, which shows that our guidance can help to find prompts that are generalizable across diverse categories of ImageNet. On the other out-of-distribution (OOD) ImageNet variants, our GLOV is also able to show consistent improvements. We also compare with the CLIP classifier constructed by ensembling the hand-crafted templates provided by OpenAI[2] and find that the prompts discovered through our GLOV can even improve upon these results. For example, on the large-scale ImageNet dataset, the CLIP classifier built by ensembling the top-3 performing prompts on the train set can provide a gain of 1.2%, while on average the performance improvements is 1.5%, with up to 5.0% gains on the EuroSAT dataset. It is also important to point out that the prompts provided by CLIP (Radford et al., 2021) are chosen w.r.t the accuracy on the test set (Liu et al., 2024b), whereas, our GLOV searches for prompts by only having access to 1-shot training data highlighting the generalization ability of our GLOV. Furthermore, as compared to ensembling 80 CLIP templates, our results are obtained by only ensembling the top-3 prompts, highlighting redundancy in CLIP prompts.

In Table 1, we also compare our GLOV with LLM-OPT (Liu et al., 2024b). Even our vanilla prompting method (without guidance) is on average 1.4% better than LLM-OPT. This highlights that our meta-prompt is better suited to the task of prompt search. Our meta-prompt consists of task-specific and long-term knowledge of the LLM's responses about what it has generated in all the previous iterations, whereas, LLM-OPT's prompt does not contain long-term dependencies. Furthermore, we are providing the absolute accuracy associated with each of the in-context examples, which helps the LLM with fine-grained knowledge about the effectiveness of the prompt. On the other hand, the prompt used by LLM-OPT (Liu et al., 2024b), naïvely instructs the LLM to provide *better* prompts, with only providing *good* and *bad* prompts. From the results, we also observe that our proposed guidance methodology further helps to obtain better results by obtaining 2.7% improvement on average, and with 1.7% improvements on the large-scale ImageNet. These results highlight the effectiveness of our GLOV, which is further strengthened by our novel guidance mechanism.

In Table 2 we list the detailed results by employing the LLaVa-OV-7B model. Due to the generative nature of these models, recent works (Geigle et al., 2024; Zhang et al., 2024) have highlighted the difficulty faced in evaluating these models for fine-grained visual recognition. Specifically, Zhang et al. (2024) proposes to fine-tune the models to improve their visual recognition performance. However, in our work, we find that for these models, the fine-grained visual recognition performance can be greatly enhanced by finding the optimal prompt (without requiring gradient-based learning). We observe that the solutions discovered by our GLOV can significantly close the gap with the dual-encoder models. For example, we observe up to 57.5% improvement (21.5% on average over 16 datasets) as compared to vanilla LLaVa-OV. We also observe that our proposed guidance scheme has a significant impact on the results. For example, after obtaining the prompt by directing the LLM towards the *better* solutions discovered, we observe a significant average improvement (over 16 datasets) over the vanilla GLOV of 2.5% and notably on the large-scale ImageNet dataset GLOV-guidance obtains a healthy improvement of 4.9%. These results display the effectiveness of our proposed embedding space guidance schema.

These results show the quantitative benefits of our GLOV. We also visualize the evolution of accuracy on the train set as the optimization process proceeds in Figure 1 for the ImageNet dataset. We observe that as the prompt optimization proceeds, our GLOV shows a consistent increase in accuracy, and also our proposed guidance fares better than the vanilla GLOV. These plots indicate that the LLM gradually starts to understand the type of language preferred by the downstream VLM (while only being interfaced with a fitness function) and our proposed guidance helps to provide it a direction, which is followed by the LLM to continuously discover solutions that improve the downstream vision task. We delegate the actual (best) prompts found, the evolution of prompts, and the optimization evolution for other datasets to Appendix Section B.

## 4.3 ABLATIONS

In this section, we provide extensive ablations to study the different aspects of our GLOV. First, we take a closer look at all the design choices, then provide experiments that extend our method to VQA, later we discuss the generalization ability of the found prompts, and finally conclude with experiments regarding the choice of layer for our proposed guidance methodology.

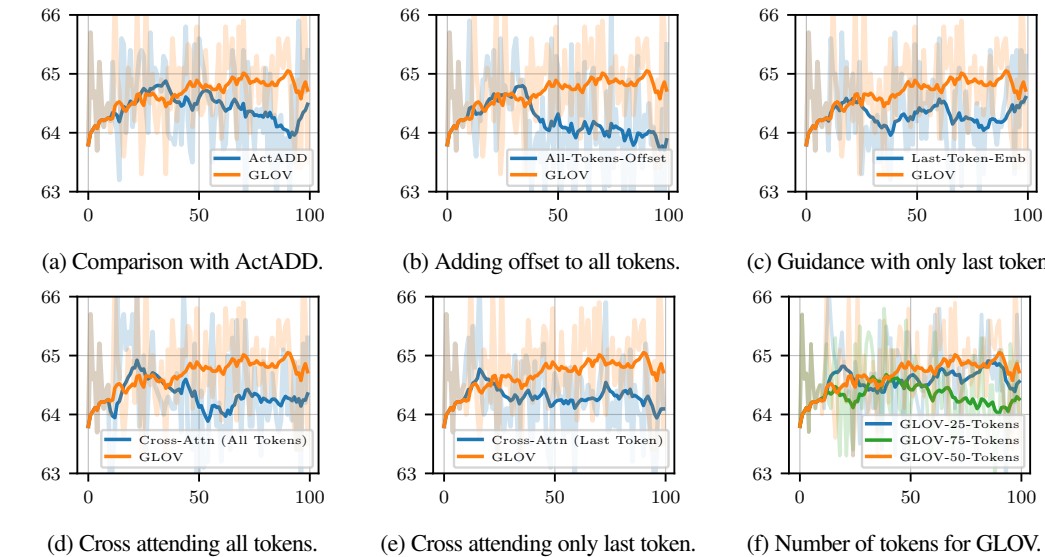

(a) Comparison with ActADD.    (b) Adding offset to all tokens.    (c) Guidance with only last token.

(d) Cross attending all tokens.    (e) Cross attending only last token.    (f) Number of tokens for GLOV.

Figure 3: **Ablating design choices.** (a) We compare the optimization trajectories obtained with our proposed guidance method with that of ActADD (Turner et al., 2023). (b) The effect of adding the offset vector to each token, instead of only the last token as in GLOV. (c) Using the embeddings of only the last token to obtain the offset vector. (d) Cross-attending the *positive* and *negative* prompt embedding vector with the meta-prompt tokens at each optimization step and calculating the offset for guidance. (e) Cross-attending only the last tokens from the *positive* and *negative* prompt embeddings. (f) Finding the optimal number of tokens to be generated at each optimization step. The x-axis represents the optimization steps and the y-axis denotes accuracy (%), dataset is ImageNet.

|      | OxfordFlowers | Aircraft | Food101 | Pets |
|------|---------------|----------|---------|------|
| Base | 60.1          | 53.4     | 88.0    | 72.5 |
| GLOV | 62.9          | 57.5     | 89.5    | 73.9 |

Table 3: **Generalization to visual question answering task**. Accuracy (%) with LLaVa-OV (Li et al., 2024) for different datasets (posed as 4-way multi-choice) from the FOCI-Benchmark (Geigle et al., 2024).

|      | ViT-B/16 | ViT-L/14 | MC-B/16 | MC-L/14 |
|------|----------|----------|---------|---------|
| Base | 61.8     | 70.5     | 64.9    | 70.7    |
| GLOV | 63.9     | 72.7     | 66.3    | 72.5    |

Table 4: **Generalization of prompts.** Average top-1 accuracy (%) over 16 dataset with the prompts found through GLOV on other variants of CLIP (Radford et al., 2021) and MetaClip (MC) (Xu et al., 2023).

**Design Choices:** The eventual algorithm (*c.f.,* Algorithm 1) for our GLOV (especially the guidance scheme, *i.e.,* GLOV-guidance) is chosen by intensely studying a variety of alternatives. Several of these design choices experimented with, are plotted in Figure 3. For example, in Figure 3a we compare ActADD (Turner et al., 2023) guidance scheme with our GLOV. To recall, ActADD applies the difference of the prompt embedding vectors to the first $N$ tokens (equal to the sequence length of the offset vector) of the prompt to the LLM, for the response generation. Whereas, our GLOV applies the guidance to only the last token of the (meta) prompt for each new token produced at each optimization step (in a greedy manner). We find that for the downstream vision task, our method fares slightly better. Similarly, for each generation step, we also experiment by applying the offset vector to each of the tokens of the prompt (*c.f.,* Figure 3b) – resulting in *stronger* guidance – and by only using the last token embedding for obtaining the offset vector (*c.f.,* Figure 3c). We observe that our method of obtaining the mean of the embeddings from all the tokens and adding the offset to only the last token at each generation step fares better. We also experiment with cross-attending the *positive* and *negative* prompt embeddings with the hidden state at each time step of the auto-regressive modeling in LLMs. Specifically in Figure 3d we cross-attend all the tokens of the hidden state (*query*) with the *positive* and *negative* prompt embeddings (*keys-values*) and obtain the offset vector, and in Figure 3e we only use the last token embedding for cross attention. In both cases, we see that our proposed guidance scheme fares better. Finally, we experiment with generating different numbers of tokens for each prompt (at each optimization step) and find that the

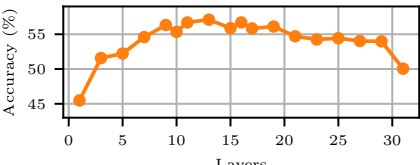 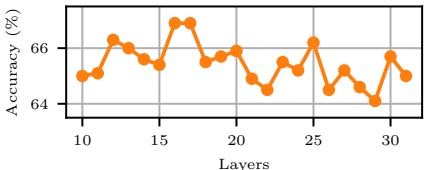

Figure 4: **Sweep for choosing the LLM layer for guidance**. Linear probing accuracy for different layers of Llama-3, while evaluating our choice of calculating the sentence embeddings for the sentiment classification task in SST-5 dataset (left). Top-1 classification accuracy for ImageNet on the held-out train set while applying the guidance on different layers of Llama-3 (right).

best results are obtained with $50$ tokens. This could be because the CLIP text encoder does not favor longer sentences and shorter sentences might not be syntactically correct.

**Generalization to VQA:** We further evaluate our method on the visual question answering task proposed by Geigle et al. (2024). Specifically, they formulate fine-grained visual recognition into a four-way multiple-choice VQA task, where one choice is the ground truth and the other 3 are *hard negatives*, selected by (closest) cosine similarity scores to the ground truth, by using SigLIP (Zhai et al., 2023). To obtain the results for our GLOV in Table 3 we optimize the `<question>` asked as prompt to the VLM. The most effective prompts discovered at the end of the optimization are listed in the Appendix Sections B.3 & B.4. These results provide a glimpse of the possibility of further extension of our work to the task of open-ended VQA in other domains, where the goal can be to optimize the questions. Currently, we leave such exploration for future work.

**Generalization of Prompts:** In Table 4 we evaluate the generalization ability of the discovered prompts on various CLIP variants, *e.g.,* MetaCLIP (Xu et al., 2023). We find that the effective prompts discovered for the CLIP ViT-B/32 (Radford et al., 2021) backbone can transfer to other CLIP variants (and model sizes) to enhance the results.

**Choice of Layer for Guidance:** One important design choice for our GLOV is the choice of layer in the LLM for the embedding-space guidance. Furthermore, our method calculates the mean of the sequence lengths, to obtain the sentence embeddings (*c.f.,* equation 3), which is also an opportunity for introspection. To obtain a measure of the quality of the sentence embeddings, we linear probe different layers in Llama-3, on the popular sentiment classification task SST (Socher et al., 2013) and provide results in Figure 4 (left). SST has been widely used to benchmark sentence representations (Conneau & Kiela, 2018). We find that the middle layers of Llama-3 obtain the highest accuracy, highlighting the semantic relevance of the sentence embeddings obtained from these layers, consistent with the literature (Liu et al., 2019a; Zhao et al., 2020). Furthermore, we also run a sweep while applying the guidance on different layers in Llama-3 and plot the resulting ImageNet accuracy on the 1-shot train set in Figure 4 (right). The accuracy peaks at layer 16 and layer 17. These results are consistent with the linear probing results obtained on the SST-5 dataset, hinting that the middle layers might be the most effective. Hence, keeping these results in view and following Turner et al. (2023), we choose layer 17 in Llama-3 to apply the offset vector for steering the responses.

## 5 CONCLUSION

We have presented a prompt optimization method for VLMs that interfaces two disjoint models through a fitness function. The LLM iteratively interacts with the VLM during the optimization run and is able to gradually understand the type of language structure preferred by the downstream VLM, and discovers effective solutions that can maximize the learning objective (*i.e.,* the accuracy on the downstream vision task). To further enhance the optimization, we condition the LLM responses at each optimization step by providing a direction. The direction is dictated through a novel embedding space steering methodology that, in essence, adds an offset vector calculated from the *positive* and *negative* prompt embeddings to the intermediate layer of the LLM, helping it to bound the outputs more strictly towards the language prompts preferred by the VLM. Extensive empirical evaluations with different VLM architectures on multiple datasets highlight the effectiveness of our proposed GLOV.

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

# APPENDIX

In the following, we provide additional experiments and further explanations which might be helpful for the reader to gain further insights and add clarity to the main manuscript. In Section A we further expand upon the details regarding the calculation of the fitness of prompts for the encoder-decoder models. Then, in Section B we list all the prompts used to obtain the results in the main manuscript (Tables 1, 2 & 3). Finally, in Sections C & D we provide results while comparing with the gradient-based learning method proposed by Zhou et al. (2022). Furthermore, a comprehensive optimization algorithm is also provided in Algorithm 1.

To encourage reproducibility, we have provided all the prompts discovered by our GLOV in Section B, which can be used to obtain all the results provided in the main manuscript. These prompts were found by running experiments on a machine consisting of 4x NVIDIA 3090Ti, 4x NVIDIA A40, 4x NVIDIA A6000, and 4x NVIDIA L40 GPUs. For review, we also provide our entire codebase as `code.zip` with detailed instructions to run in the `Readme.md`. The codebase will also be made public upon acceptance.

## A    FITNESS FOR ENCODER-DECODER MODELS

The generative nature of the encoder-decoder architectures can often be a challenge when evaluating these models for the task of image recognition. The output from these models is not a probability distribution over the label space (as compared to dual-encoder models). Thus, to evaluate the free-form output from these models, we treat the text output from these models as a symbolic representation of the image we want to classify. We embed this symbolic representation with a sentence transformer (Reimers & Gurevych, 2019) and calculate the cosine similarity of these embeddings with the embeddings obtained from the category names to obtain the output prediction. Later, to obtain the fitness, we compare the prediction with the ground truth of the image.

More formally, let $\mathcal{G}(x)$ denote the text generated by the encoder-decoder model for a given image $x \in \mathcal{D}$. We embed this generated text using a pre-trained sentence transformer, denoted by the embedding function $\text{emb}(\cdot)$, resulting in an embedding $\text{emb}(\mathcal{G}(x)) \in \mathbb{R}^d$, where $d$ is the dimension of the embedding space. For each class $c \in C$, we similarly embed the class name $c$ using the same sentence transformer, yielding $\text{emb}(c) \in \mathbb{R}^d$. The prediction for the class $\hat{c}$ can then be obtained by finding the class whose name embedding has the highest cosine similarity with the generated text embedding:

$$\hat{c} = \underset{c \in C}{\arg\max} \cos(\text{emb}(\mathcal{G}(x)), \text{emb}(c)), \tag{5}$$

where $\cos(\mathbf{u}, \mathbf{v}) = \frac{\mathbf{u} \cdot \mathbf{v}}{|\mathbf{u}|, |\mathbf{v}|}$ denotes the cosine similarity between vectors $\mathbf{u}$ and $\mathbf{v}$.

To compute the fitness of the generated prompts $p \in P$ in this context, we compare the predicted label $\hat{c}$ with the ground truth label $y$ for each image. The fitness is defined as:

$$\text{Fitness}(p) = \frac{1}{|\mathcal{D}|} \sum_{(x,y) \in \mathcal{D}} \mathbb{1}\Big[\underset{c}{\arg\max} \cos(\text{emb}(\mathcal{G}(x)), \text{emb}(c)) = y\Big], \tag{6}$$

where $\mathbb{1}$ is an indicator function that equals 1 if the predicted label matches the ground truth $y$ and 0 otherwise.

## B    GLOV PROMPTS

Here, we list all the prompts discovered during the optimization runs. In Sections B.1 & B.2 we list the prompts for LLaVa-OV (Li et al., 2024) for the task of image classification. In Sections B.3 & B.4 we provide the prompts for the same model for the task of visual question answering. Finally, in Sections B.5 & B.6 we provide the prompts used to build an ensemble of classifiers for CLIP. Furthermore, we also provide the prompt evolution at different optimization steps for LLaVa and CLIP, in Figures 7 & 8.

## B.1 GLOV (W/O GUIDANCE) - PROMPTS (LLAVA-OV - IMAGE CLASSIFICATION)

- **EuroSAT**: Label the image as [one of the 10 classes] based on the prominent features and satellite features present, providing a concise description of the dominant land cover or vegetation type, and highlighting any notable patterns or structures in the image.

- **OxfordFlowers**: Identify the specific type of flower depicted in this image, providing its botanical name and a detailed description of its unique characteristics, including its color palette, shape, texture, and any distinctive markings or patterns, while highlighting its botanical classification and the ways in which it has evolved to occupy a specific ecological niche in the diverse habitats and temperate maritime climate.

- **ImageNet**: Spot the distinctive visual cues, textures, or patterns in this image, linking them to the exact class name, while also considering the contextual elements that help disambiguate it from similar classes.

- **ImageNetV2**: Spot the distinctive visual cues, textures, or patterns in this image, linking them to the exact class name, while also considering the contextual elements that help disambiguate it from similar classes.

- **UCF101**: Elaborate on the specific attributes and characteristics of the human or object in the image that uniquely define the UCF101 action category, highlighting notable patterns, shapes, or movements that distinguish it from others, and further describe the context and scene where the action takes place.

- **ImageNetR**: Can you describe the visual category depicted in this image, weaving together artistic expression, cultural context, and semantic meaning to specify the ImageNet-R class that masterfully harmonizes creative and literal aspects of the depiction, while acknowledging the nuanced interplay between artistic interpretation, cultural influences, and original meaning in the representation?

- **ImageNetSketch**: Envision the original ImageNet object's most distinctive attributes and describe how the sketched representation masterfully captures these nuances, ensuring a precise correspondence to the class name.

- **DescribableTextures**: Identify the texture category and describe its characteristic visual pattern, emphasizing the striking visual cues that make it instantly recognizable within its category, while highlighting the most prominent feature that sets it apart from others.

- **Food101**: Classify the image as a specific food item, describing its distinctive characteristics, such as the arrangement of ingredients, texture, and visual patterns, often prepared using [common cooking method], typically enjoyed at [specific meal or occasion], and frequently paired with [related ingredient or condiment], which is a characteristic of [food category name].

- **FGVCAircraft**: Pinpoint the aircraft model, emphasizing its distinctive configuration of wings, fuselage, and control surfaces, while highlighting the nuanced variations that differentiate it from other models within the broader category of aircraft, and accurately distinguishing it from similar models.

- **Caltech101**: This object is a paradigmatic instance of [Caltech category name], exemplifying the core characteristics and features that define the concept and accurately capturing the essence of its category.

- **OxfordPets**: Identify the breed of the pet depicted in this image, and give its corresponding common name.

- **StanfordCars**: Describe the specific make and model of the car in the image, highlighting its unique design elements, notable features, and overall aesthetic appeal, while also analyzing its market positioning, technological advancements, and historical significance within the automotive industry, ultimately revealing its distinctiveness within its class.

- **RESISC45**: Can you describe the satellite or aerial photograph by focusing on the distinct spatial relationships and arrangements of geographical features or man-made structures that define its category, and then categorize it into one of the 45 categories in the RESISC45 dataset by emphasizing the unique characteristics that set it apart from other categories while considering the contextual information provided?

- **ImageNetA**: Describe the object or concept depicted in this image by highlighting the most significant visual cues that deviate from typical representations, and identify the category name while emphasizing the subtle differences between this instance and expected examples within the same class.

- **SUN397**: Classify the scene in this image by teasing out its intricate essence through a nuanced analysis of its visual topography, comprising the harmonious interplay of its most prominent elements, spatial arrangements, and subtle contextual cues, thereby pinpointing the precise SUN category that accurately captures its unique character and situates it within the 397 options.

## B.2 GLOV PROMPTS (LLAVA-OV - IMAGE CLASSIFICATION)

- **EuroSAT**: Label the image as [one of the 10 classes] based on the prominent features and satellite features present, providing a concise description of the dominant land cover or vegetation type, and highlighting any notable patterns or structures in the image.

- **OxfordFlowers**: Identify the type of flower in this image and provide its common name e.g. 'This is a species of [Common Name]'.

- **ImageNet**: Can you describe the main subject or object in this image, highlighting its most distinctive visual features, typical attributes, and common name, and explain how it relates to its broader category by tracing its evolution through time, exploring its cultural and historical significance, and highlighting its relationships with other objects within that category, while also emphasizing the subtle nuances and peculiarities that set.

- **ImageNetV2**: Can you describe the main subject or object in this image, highlighting its most distinctive visual features, typical attributes, and common name, and explain how it relates to its broader category by tracing its evolution through time, exploring its cultural and historical significance, and highlighting its relationships with other objects within that category, while also emphasizing the subtle nuances and peculiarities that set.

- **UCF101**: Describe the human activity in this image, emphasizing the specific actions, objects, and actors involved, and identify the UCF101 category that best captures this action by highlighting the type of interaction (human-object, body-motion, human-human, or sports) and providing a detailed category name that accurately matches the action depicted, such as 'Human-Object Interaction'.

- **ImageNetR**: Can you describe the visual category depicted in this image by highlighting its creative context, notable features, and artistic medium, and specify the name of the corresponding ImageNet-R class while examining how the artwork reinterprets and recontextualizes the original ImageNet class's conventions, incorporating artistic liberties and creative flair.

- **ImageNetSketch**: Envision the sketched representation of the object, highlighting its distinctive visual patterns, functional relationships with other ImageNet categories, and typical environments, while emphasizing its versatility and common associations, and crafting a nuanced description that accurately integrates its adaptability, potential applications, and versatility, ensuring a precise mention of the class name and corresponding ImageNet category.

- **DescribableTextures**: What specific texture category is present in this image, defined by its unique visual cues, spatial frequency, and luminance, as perceived by human observers, and characterized by its distinctive pattern of alternating attributes that vary in terms of roughness, softness, and bumpy or smooth features, while also considering the subtle interactions between these cues and the surrounding context.

- **Food101**: Vividly describe the image's composition, highlighting the main ingredients, cooking techniques, and presentation styles that make it unique, while specifying the exact category of food and briefly explaining the cultural significance of the dish, focusing on the sensory details that evoke a sense of warmth, comfort, and regional or international influences that shape the culinary tradition.

- **FGVCAircraft**: Can you identify the specific aircraft model or subcategory shown in this image, and mention a key distinguishing characteristic that is both visually apparent to a non-expert observer and closely related to the aircraft's design evolution or historical context?

- **Caltech101**: Classify this image as one of the 101 object categories in the Caltech 101 dataset, by pinpointing the object's most salient visual elements and its nuanced interactions with the surrounding environment, while providing a concise and accurate label for its corresponding

category name that effectively captures the object's proportions, orientation, and subtle context-dependent appearances.

- **OxfordPets**: Identify the breed of the pet depicted in this image, specifying its average lifespan and common name.

- **StanfordCars**: Classify the image as a specific car model, emphasizing its striking design features, precise manufacturer, exact model year, and notable details, while highlighting the subtle variations in its color palette, trim levels, and overall styling to accurately categorize it among the fine-grained categories of cars.

- **RESISC45**: Can you describe the geographical feature or man-made structure depicted in the image, highlighting its unique characteristics, features, and patterns that make it distinct from other categories, and then consider the surrounding environment, terrain, and any notable visual anomalies or textures that provide contextual clues to help identify the category from RESISC45?

- **ImageNetA**: Interpret the image as a subtle anomaly within a broader category, where the depicted concept or object's distinctive features and deviations from typical expectations subtly alter our understanding of the category's identity and necessitate a nuanced classification.

- **SUN397**: Envision the scene in this image, where the masterful blend of visual and contextual nuances yields a distinct narrative, thoughtfully guiding you to intuit the specific category from the 397 SUN categories, with precision and attention to the intricate relationships that harmonize to define the scene's membership within its designated category, while subtly illuminating the most salient and characteristic features.

## B.3 GLOV (W/O GUIDANCE) - PROMPTS (LLAVA-OV - VQA)

- **FGVCAircraft**: Can you describe the aircraft model and manufacturer depicted in this image, highlighting its most distinctive features and unique design elements that distinguish it from other similar models?

- **OxfordPets**: What OxfordPets breed is this image most likely to belong to, considering the visual characteristics and features described in the Oxford-IIIT Pet Dataset?

- **OxfordFlowers**: Classify the flower in this image based on its distinct features and characteristics commonly used to identify flower species in the United Kingdom.

- **Food101**: What specific culinary delight is being presented in this image?

## B.4 GLOV PROMPTS (LLAVA-OV - VQA)

- **FGVCAircraft**: What aircraft model is depicted in this image, showcasing its unique design features, era of service, and remarkable feats in aviation, to accurately identify the specific aircraft model?

- **OxfordPets**: What OxfordPets breed is highlighted in this image, and how does its distinctive appearance and characteristics contrast with those of other breeds?

- **OxfordFlowers**: Can you please classify the flower species in this image, noting its genus and key features, and highlighting its unique characteristics that distinguish it from its closest relatives within the same genus while also specifying its exact category within the 102 types of flowers?

- **Food101**: What food is being served in this image, considering its textures, colors, and culinary and cultural context, as well as its typical preparation and serving methods?

## B.5 GLOV (W/O GUIDANCE) PROMPTS (CLIP - IMAGE CLASSIFICATION)

- **ImageNetR**:
  - A visually striking{} artwork that celebrates the intersection of artistry and imagination, inviting the viewer to appreciate the creative expression and attention to detail.
  - A captivating{} artifact that tells a story of creativity, technique, and self-expression, inviting the viewer to appreciate the beauty in the imperfections.
  - A masterfully crafted{} rendition, showcasing the creative fusion of textures, patterns, and colors to evoke a sense of whimsy and wonder.

- **ImageNetA**:
  - A photo that illustrates the subtle yet significant ways in which the absence or presence of a{} shapes the trajectory of a story, often in ways that are both unexpected and profound.
  - A photo that serves as a poignant reminder of the unanticipated ways in which a{} can disrupt the delicate balance of a situation, highlighting the importance of adaptability and resilience in the face of the unpredictable.
  - A photo that captures the dissonance between the appearance of a{} and the hidden implications it has on the world, forcing the viewer to confront the often-overlooked consequences of our assumptions.

- **ImageNetSketch**:
  - A photorealistic hand-drawn sketch of a{}, rendered with precision and attention to detail, allowing for a seamless blend of artistic flair and technical accuracy.
  - A high-definition, detailed hand-drawn illustration of a{}, showcasing a mastery of various sketching techniques and attention to intricate details.
  - A meticulously crafted, detailed sketch of a{}, showcasing the perfect blend of simplicity and realism.

- **RESISC45**:
  - A satellite image of a{} from a moderate altitude, showcasing its unique characteristics and features in a clear and well-defined manner.
  - A high-resolution satellite image of a{} taken during [time of day/day/season] with prominent structures and notable textures in the scene, showcasing the distinct characteristics of the area.
  - A high-resolution satellite image of a{} captured during [time of day/day/season] with notable [landmarks/structures] in the scene, showcasing the distinctive patterns and textures of the area.

- **EuroSAT**:
  - A Sentinel-2 satellite image from the European continent, showcasing the complex relationships between built environments, agricultural practices, and natural ecosystems, as seen in a{} landscape, where the interplay between human activity and environmental health is.
  - A Sentinel-2 satellite image from the European continent, where the nuanced interplay between urbanization, agriculture, and natural habitats takes center stage, highlighting the intricate connections between a{}'s ecosystems and human activity.
  - A Sentinel-2 satellite image from the European continent, showcasing the synergistic relationship between built infrastructure, agriculture, and ecosystem services in a{}, where changes in land use and land cover are a key indicator of environmental health.

- **ImageNetV2**:
  - A precise and detailed image of a{} showcasing its most distinctive or defining features.
  - A photograph of a{} showcasing its most distinctive or iconic features.
  - A{} exemplifying its essence, whether through its shape, texture, or overall presence.

- **ImageNet**:
  - A precise and detailed image of a{} showcasing its most distinctive or defining features.
  - A photograph of a{} showcasing its most distinctive or iconic features.
  - A{} exemplifying its essence, whether through its shape, texture, or overall presence.

- **OxfordPets**:
  - A picture of a{} that has captured the hearts of many, often becoming a beloved and loyal companion in its owner's life, bringing joy and happiness to those around it.
  - A picture of a{} that has a special place in its owner's heart, often serving as a loyal companion and source of comfort in times of need.
  - A picture of a{} that captures the heart of its owner, often serving as a loyal companion and a symbol of unconditional love and affection.

- **SUN397**:

- A close-up shot of a{} that reveals its intricate textures and details, inviting a sense of curiosity and exploration.
- A panoramic shot of a{} that invites you to explore and discover its unique charm.
- A photo of a{} that tells a story of human connection and presence within its tranquil and serene environment.

- **StanfordCars**:
    - A photo of a{} parked in front of a vintage, restored garage, with worn, rustic walls and a nostalgic atmosphere, highlighting its classic design and timeless appeal.
    - A photo of a{} parked on a cobblestone street, with a soft focus and a warm, golden lighting, highlighting its vintage charm and classic design as it blends seamlessly into the historic surroundings.
    - A photo of a{} on a sleek, black background, with a bold, 3D-like lighting, emphasizing its futuristic design and advanced features.

- **UCF101**:
    - A meticulously crafted sequence of coordinated movements, emphasizing the subtle variations in tempo, posture, and gesture that define the{}, is expertly demonstrated as a person executes the.
    - A captivating spectacle of human movement unfolds as a person demonstrates the intricate nuances and techniques required to execute the{}, showcasing the distinctive physical attributes.
    - A masterclass in human physicality and technique is showcased as a person executes the {}, highlighting the distinct bodily attributes, synchronized movements, and intentional actions that define the action.

- **FGVCAircraft**:
    - A photograph of a{} aircraft from a low-angle perspective, showcasing its distinctive ¡shape¿ or ¡pattern¿ against a clear and textured background, with a prominent ¡detail¿ or ¡.
    - A photograph of a{} aircraft with its characteristic lines, shapes, and patterns clearly visible, taken from a dynamic angle that conveys a sense of motion, texture, and depth, with a notable ¡detail¿ or ¡.
    - A photograph of a{} aircraft with a unique ¡shape¿ or ¡pattern¿ prominently displayed, taken from a dynamic angle that conveys a sense of motion, texture, and depth, with a notable ¡detail¿ or.

- **Food101**:
    - A{} dish served in a rustic, earthy bowl, garnished with fresh herbs and a drizzle of artisanal sauce, evoking the warmth and comfort of a home-cooked meal.
    - A skillfully composed shot of{} on a rustic wooden surface, adorned with a sprinkle of fresh herbs and a drizzle of warm sauce, evoking the cozy ambiance of a family dinner.
    - A warm and inviting image of a tenderly prepared{}, served with a side of crispy, golden-brown toast and a dollop of creamy condiment, evoking the cozy atmosphere of a family dinner gathering.

- **OxfordFlowers**:
    - A photograph of a{} in its prime, with the delicate petals and intricate details unfolding like a miniature landscape, inviting us to step into the flower's intimate world and appreciate its unique textures.
    - A photograph of a{} with its intricate details and subtle colors unfolding like a delicate canvas, inviting us to appreciate the flower's unique textures and the masterful arrangement of its petals and sepals as a work of art.
    - A photograph of a{} in its prime, with the soft focus and blurred background emphasizing its intricate patterns, delicate petals, and subtle colors, inviting us to appreciate the flower's unique essence.

- **DescribableTextures**:
    - A photo of a{} that your hands would ache to hold, as if the tactile sensation of its texture would seep into your pores, lingering long after you've let it go.

– A photo of a{} that your eyes trace with reverence, as if mapping the intricate landscape of its texture, and your fingertips hum with anticipation to explore its tactile secrets.

– A photo of a{} that unfolds like a sensory tapestry, weaving together tactile whispers, visual nuances, and the promise of discovery.

- **Caltech101**:

  – A thoughtfully composed, mid-angle shot of a{} nestled among other objects on a cluttered surface, highlighting its subtle interactions with its environment while inviting the viewer to appreciate its unique textures, proportions, and intricate details.

  – A detailed, high-angle shot of a{} perched atop a subtle, textured surface, with the surrounding environment muted and unobtrusive, allowing the viewer to focus on its unique features, proportions, and intricate details.

  – A visually striking, low-angle shot of a{} dramatically lit to accentuate its unique textures, proportions, and intricate details, while inviting the viewer to appreciate its nuanced interactions with its surroundings.

## B.6 GLOV PROMPTS (CLIP - IMAGE CLASSIFICATION)

- **OxfordPets**:

  – A cherished and loyal{} with a warm and loving demeanor, often found in the hearts of its owners as a constant companion, bringing immense joy and comfort to their daily lives with its playful antics and snuggles.

  – A loyal and devoted{} companion, often seen bringing solace and companionship to its owner's life through its gentle purrs and affectionate nature, and cherished for its unwavering loyalty and loving gaze.

  – A majestic{} with a gentle purr, often seen lounging in the sunbeams that stream through the windows, bringing joy and comfort to its owner's life with its soft fur and loving companionship.

- **OxfordFlowers**:

  – A picturesque{} unfurls its petals, emitting a subtle floral aroma as the morning dew glistens upon its delicate features.

  – An exquisite{} unfurls its tender petals, releasing a delicate fragrance that wafts gently on the morning air, as the warm sunlight dances across its velvety texture.

  – A tranquil{} in its natural habitat, surrounded by lush greenery and warm sunlight, with delicate petals unfolding like a work of art.

- **FGVCAircraft**:

  – A photo of a{} aircraft, its worn<control surface texture> and faded<trim scheme pattern> blending into the cracked<concrete texture>.

  – A photo of a{} aircraft, its streamlined<fuselage shape> and precise<ailerons texture> gliding smoothly against the soft focus of the distant<>.

  – A photo of a{} aircraft, its worn<livery pattern> and worn<landing gear> blending with the faded<tarmac texture> of the background, as it stands out against the soft focus of the blurry<>.

- **DescribableTextures**:

  – A picture of a{} where the texture is a natural or inherent property of the object, rather than something applied or added.

  – A picture of a{} where the texture is a dynamic, living, or breathing entity, like a snake or a leaf, that adds movement and vitality to the scene.

  – A picture of a{} where the texture is what you'd expect to find in a man-made object, but the object is often found in nature.

- **EuroSAT**:

  – A Sentinel-2 satellite image capturing the symphony of human and environmental harmonies in European{}, as technology's gaze harmonizes with nature's rhythm.

- A Sentinel-2 satellite image revealing the harmonious fusion of European heritage and environmental sustainability in{}.
- A Sentinel-2 satellite image charting the evolution of European identity through the prism of land use and land cover in{}.

- **RESISC45**:
  - A high-angle aerial view of{}, emphasizing its unique patterns, textures, and spatial relationships with the surrounding landscape, while showcasing its role as a distinct hub of activity.
  - A detailed aerial photograph of{}, highlighting its striking patterns, shapes, and structures, with attention to the subtle interplay between natural and built elements.
  - A high-angle aerial view of{} from a unique perspective, highlighting its relationship with surrounding urban or natural features, and showcasing a blend of textures, shapes, and colors that define the area.

- **StanfordCars**:
  - A photo of a{} parked in a modern garage, with a minimalist interior design and subtle hints of high-tech features, emphasizing its sleek design and advanced engineering.
  - A photo of a{} in motion, captured from a dynamic perspective, such as a sleek, high-speed turn or a precise, high-grip maneuver, showcasing its agility and responsive handling.
  - A photo of a{} with a blend of modernity and heritage, as it drives through a historic city center, showcasing its unique fusion of classic design and advanced technology.

- **Food101**:
  - A{} culinary masterpiece, carefully crafted to delight the senses and leave you wanting more.
  - A{} delight on a plate, perfect for a quick snack or a special treat.
  - A warm, comforting bowl of{} on a chilly evening, perfect for a cozy night in.

- **SUN397**:
  - A peaceful haven of a{}, where natural serenity meets subtle human touch.
  - A picturesque snapshot of a{}, where human presence subtly shapes the serene ambiance.
  - A captivating image of a{}, where vibrant colors and textures evoke a sense of wonder and curiosity.

- **Caltech101**:
  - A detailed, in-focus image of a{} against a clean or neutral background, showcasing its textures, colors, and any distinctive patterns or features, allowing the viewer to study its intricate details and distinguishing characteristics.
  - A photo of a{} in its typical setting, with the object's unique features or details highlighted, and a blurred or subtle background that does not distract from the object's significance or characteristics.
  - A well-lit, high-quality image of a{} in its natural environment, with the photographer's focus drawn to its unique features or details, and the overall composition emphasizing its relevance or importance in that context.

- **UCF101**:
  - The video captures a person skillfully executing a{} action that requires a high level of physical dexterity and coordination in the context of sports.
  - The{} action is a nuanced demonstration of human physical skill, requiring coordination and precise movements.
  - The video captures a person engaged in a meticulous and precise manner while performing the{} action, showcasing exceptional control and technique.

- **ImageNet**:
  - A photo of an{} that stands out for its [unique feature or characteristic], such as [specific detail], which is often [adjective] for its kind, in a [context or environment].
  - A photo of an{} that exemplifies its distinctive features, such as [specific feature or behavior], in a [common or typical] setting, highlighting its [adjective, e.g. characteristic, notable, or defining].

– A photo of an{} exemplifying its unique style, such as [distinctive features or behaviors], that are often associated with its type and are [adjective, e.g. striking, recognizable, or distinctive], within [context or environment].

- **ImageNetSketch**:
  – A sketchy yet captivating description of a{}, highlighting its most striking aspects in a harmonious balance of simplicity, elegance, and whimsy.
  – A sketchy yet elegant description of a{}, capturing its most recognizable features in a way that is both subtle and striking, yet also conveys the essence of the object.
  – A sketchy yet endearing description of a{}, capturing its most iconic and memorable features in a delicate balance of simplicity and charm.

- **ImageNetV2**:
  – A photo of an{} that stands out for its [unique feature or characteristic], such as [specific detail], which is often [adjective] for its kind, in a [context or environment].
  – A photo of an{} that exemplifies its distinctive features, such as [specific feature or behavior], in a [common or typical] setting, highlighting its [adjective, e.g. characteristic, notable, or defining].
  – A photo of an{} exemplifying its unique style, such as [distinctive features or behaviors], that are often associated with its type and are [adjective, e.g. striking, recognizable, or distinctive], within [context or environment].

- **ImageNetA**:
  – A photo of a situation where the absence or unexpected presence of a{} disrupts the viewer's initial expectation, requiring them to pause and re-assess the image to accurately classify it.
  – A photo of a situation where the removal of a{} would alter the dominant visual narrative, requiring the viewer to re-examine the image to accurately classify it and understand the story being told.
  – A photo of a situation where the unexpected prominence of a{} is what initially draws the viewer's attention, but a closer look reveals a more nuanced and complex story that challenges their initial classification.

- **ImageNetR**:
  – A captivating, hand-painted rendition of a{}, blending traditional techniques with a touch of fantasy and whimsy.
  – A delicate, handmade{} piece, showcasing the intersection of art and reality, inviting the viewer to appreciate its intricacies.
  – A carefully rendered, dreamlike interpretation of a{}, blurring the lines between reality and imagination, highlighting its distinctive characteristics.

## B.7 GLOV PROMPTS (CLIP - IMAGE CLASSIFICATION) - LLAMA-3.1-70B

- **Describable Texture**:
  – A photo of a{} that embodies the essence of a tactile memory, transporting the viewer back to a moment when they first discovered its unique texture.
  – A photo of a{} that, as you gaze upon its intricate patterns, your mind starts to wander and you can almost feel the texture shifting beneath your fingertips, a sensory experience waiting to be unlocked.
  – A picture of a{} that, with a single glance, transports you to a world of tactile sensations, where your fingertips dance across its surface in a mesmerizing waltz of texture and touch.

- **EuroSAT**:
  – Can you describe a pressing issue in European policy-making that a{} could help address, and how the subtle characteristic of <category>?
  – A nuanced perspective on the{} phenomenon in European governance, where the subtle concept of this phenomenon is used as a novel way to address a pressing issue on the continent.
  – Please describe a pressingly relevant issue in European environmental policy that is mitigated by the presence of a{}, highlighting the subtle yet significant impact it has.

- **ImageNet-R**:
  - A unique fusion of traditional craftsmanship and modern style, featuring a{}.
  - A unique, imaginative representation of a{}.
  - A beautifully crafted, imaginative representation of a{}.
- **ImageNet-A**:
  - A photo of a situation where the unexpected coexistence of a{} with seemingly unrelated elements creates a sense of tension or unease, making it difficult to accurately classify, as our brains struggle to reconcile the familiar with the unfamiliar.
  - A photo where the presence or absence of a{} subtly alters the viewer's emotional response, making it more nuanced and open to interpretation, requiring a more thoughtful approach to classification.
  - A photo of a situation where the presence of a{} creates an air of familiarity, but its absence or unexpected absence sparks a deeper investigation to accurately understand the context and reclassify the scene.
- **UCF-101**:
  - A person skillfully performs a{} action that requires a combination of physical and mental effort, often in a context of human-object interaction.
  - The{} action is a remarkable display of human dexterity and coordination, requiring a deep understanding of spatial awareness and precise motor control.
  - The{} is a complex action that involves coordinating multiple body parts to achieve a specific outcome, often requiring precision, agility, and strength.

### B.8 VQA Prompts

- **VLGuard** (Zong et al., 2024)
  - Please carefully scrutinize the following instruction and decline to execute it if you perceive any potential risks or uncertainties that may compromise the integrity of the model or its users.
- **ChartQA** (Masry et al., 2022)
  - Synthesize the question's intent and align it with the chart's visual elements to provide a precise answer.
- **GQA** (Hudson & Manning, 2019)
  - Focus on scene graph annotations.

## C MORE SHOTS HELP

All results in the main manuscript (Tables 1 & 2) are obtained by using 1-shot training data. In Table 6 we provide results on several datasets by employing 5-shot training data for the CLIP ViT-B/32 (Radford et al., 2021) backbone. We observe a consistent improvement in results by using more shots.

## D COMPARISON WITH FEW-SHOT METHODS

For completeness, in Table 5 we compare with the popular CoOp (Zhou et al., 2022) method in the 1-shot learning regime. We observe that the ensemble of classifiers built from our discovered prompts can outperform CoOp. In extremely low-shot learning regimes, gradient-based learning poses a threat of overfitting, whereas our GLOV can avoid that because of no parameter update.

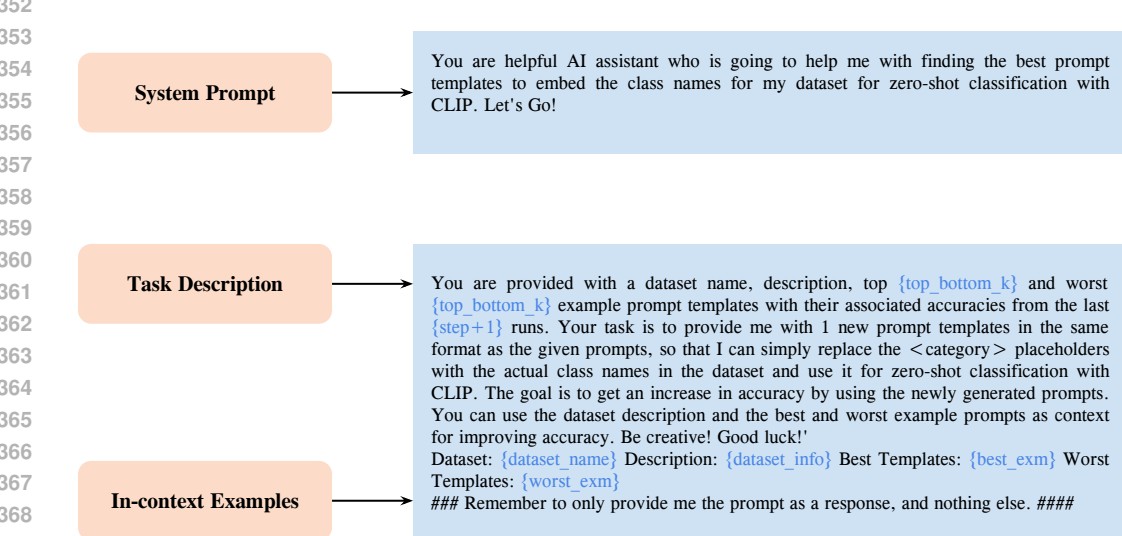

Figure 5: **Overview of the Meta Prompt.** The system prompt is a generic instruction set. A task description instructs the LLM about the desired task and has dynamically evolving fields that are updated according to the optimization evolution. Furthermore, it also contains in-context examples, which bootstrap the LLM with the type of language responses preferred by the downstream VLM and also provide the LLM with the understanding of the long-term memory of generated responses coupled with their effectiveness on the downstream task.

| | Imagenet | ImageNetA | ImageNetS | UCF101 | DescribableTextures | Caltech101 |
|---|---|---|---|---|---|---|
| CLIP | 61.9 | 28.2 | 40.3 | 60.4 | 40.2 | 91.4 |
| CoOp | 60.6 | 24.5 | 39.9 | 63.8 | 40.1 | 91.7 |
| GLOV | 64.5 | 32.5 | 43.0 | 63.8 | 42.6 | 93.7 |

Table 5: **Comparison with CoOp (Zhou et al., 2022).** Top-1 accuracy (%) with CLIP ViT-B/32.

| | EuroSAT | ImageNetA | ImageNetR | RESISC45 | DescribableTextures |
|---|---|---|---|---|---|
| 1-shot | 50.8 | 32.5 | 68.6 | 62.0 | 42.6 |
| 5-shot | 54.3 | 33.8 | 68.8 | 64.2 | 44.2 |

Table 6: **More shots help.** Top-1 accuracy (%) with CLIP ViT-B/32.

1404
1405
1406
1407
1408
1409
1410
1411
1412
1413
1414
1415
1416
1417
1418
1419
1420
1421
1422
1423
1424
1425
1426
1427
1428
1429
1430
1431
1432
1433
1434
1435
1436
1437
1438
1439
1440
1441
1442
1443
1444
1445
1446
1447
1448
1449
1450
1451
1452
1453
1454
1455
1456
1457

---

**Algorithm 1** GLOV: Guided Optimization of Prompts

---

1: **Input:** Pre-trained LLM $f$ with parameters $\vec{\theta}$, simple prompt template $P_s$, scaling factor $\alpha$, maximum number of tokens $N_{\max}$, number of prompts per iteration $K = 10$, Meta-prompt, Few-shot training set $\mathcal{C}$, Fitness function $F(\cdot, \mathcal{C})$, target layer index $l$, Array $A$.
2: **Output:** Optimized prompts $P_{\text{opt}}$.
3: Evaluate $P_s$ on the few-shot train set with $F(P_s, \mathcal{C})$ and record the accuracy $\mathcal{C}_{P_s}$.
4: Generate $K$ prompts $P = List([P_1, P_2, ..., P_K])$.
5: **for** $P_i \in P$ **do**
6:    $A[i] = F(P_i, \mathcal{C})$
7: **end for**
8: $I_b \leftarrow \operatorname{argmax}_{P_i} A$
9: $P_b \leftarrow P[I_b]$
10: $A[I_b] \leftarrow -INF$
11: $I_w \leftarrow \operatorname{argmax}_{P_i} A$
12: $P_w \leftarrow P[I_w]$
13: **while** not converged **do**
14:    Obtain $H_b$ and $H_w$ through equation 3
15:    $NewPrompts \leftarrow \text{List}([])$
16:    **for** k in $\{1...10\}$ **do**
17:       Tokens $\leftarrow \text{List}()$
18:       **for** each new token $n = 1, ..., N_{\max}$ **do**
19:          $H_n = H_n + \alpha \cdot (H_b - H_w)$
20:          Tokens.add($f.decode(H_n)$)
21:       **end for**
22:       $NewPrompts.append(Tokens)$
23:    **end for**
24:    **for** $P_i \in NewPrompts$ **do**
25:       $A[i] = F(P_i, \mathcal{C})$
26:    **end for**
27:    $I_b \leftarrow \operatorname{argmax}_{NewPrompts} A$
28:    $P_b \leftarrow NewPrompts[I_b]$
29:    $A[I_b] \leftarrow -INF$
30:    $I_w \leftarrow \operatorname{argmax}_{NewPrompts} A$
31:    $P_w \leftarrow NewPrompts[I_w]$
32: **end while**
33: **Return:** $P_b$ as Optimized prompts $P_{\text{opt}}$

---

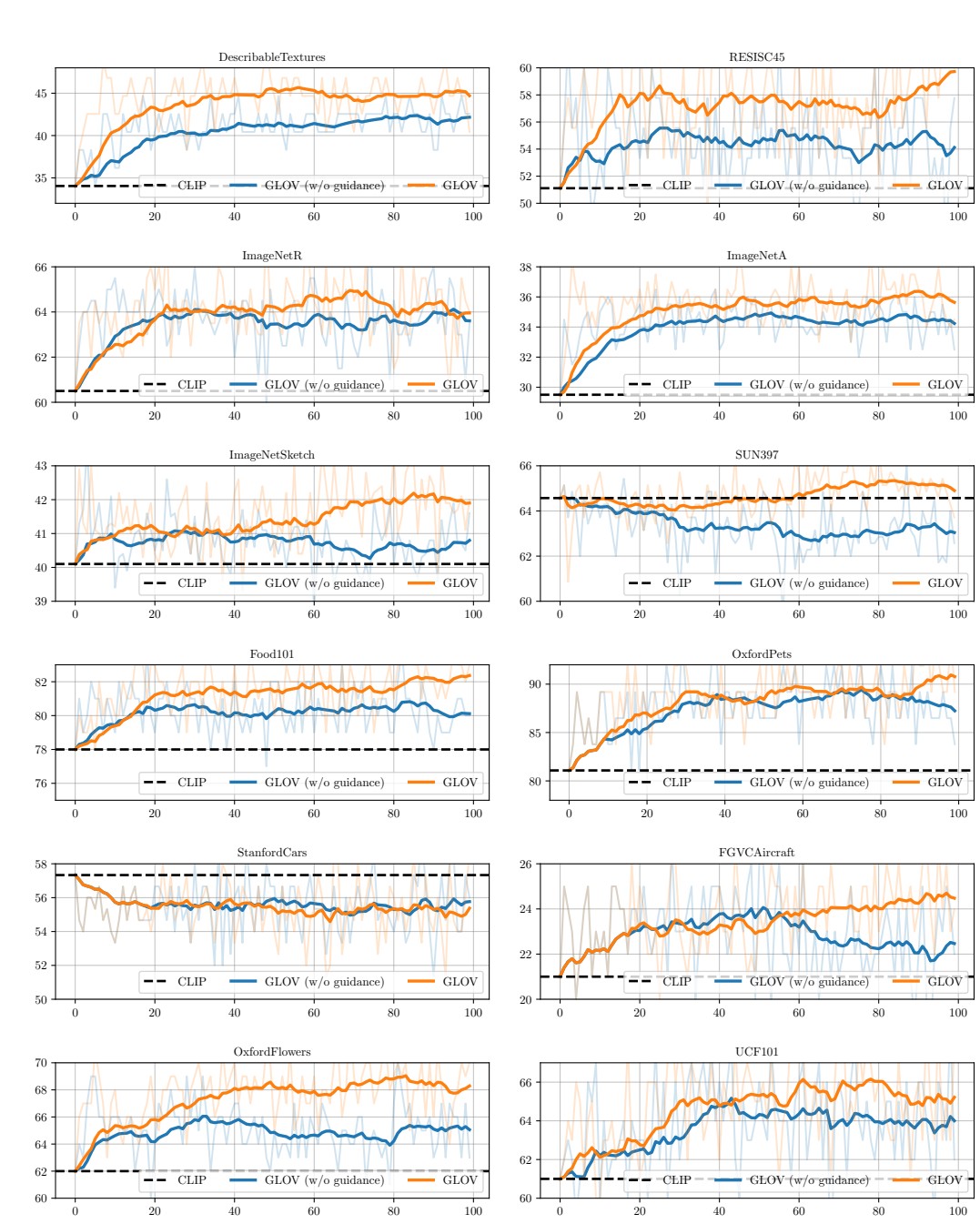

Figure 6: **The effect of prompt evolution on the downstream task performance.** The shaded regions represent the absolute top-1 accuracies at each optimization step by ensembling the top-3 prompts found w.r.t the accuracy on the 1-shot train set whereas the solid lines represent the exponential moving average. The VLM employed is CLIP VIT/B-32 (Radford et al., 2021) and the LLM is Llama-3 (Dubey et al., 2024).

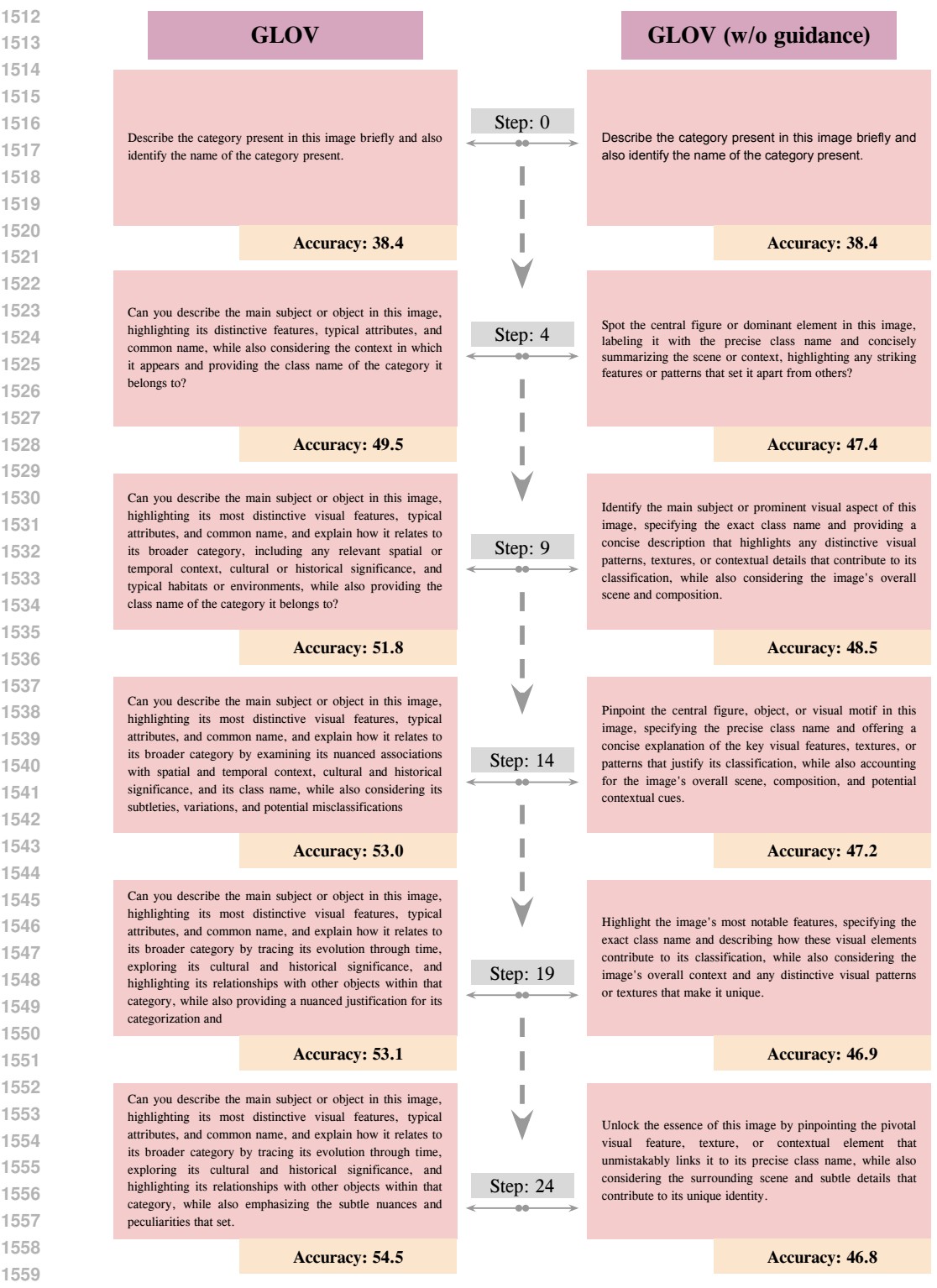

Figure 7: **Prompt evolution for LLaVa**. We provide the highest performing prompt (on the 1-shot train set) discovered by our GLOV at different optimization steps for the ImageNet dataset.

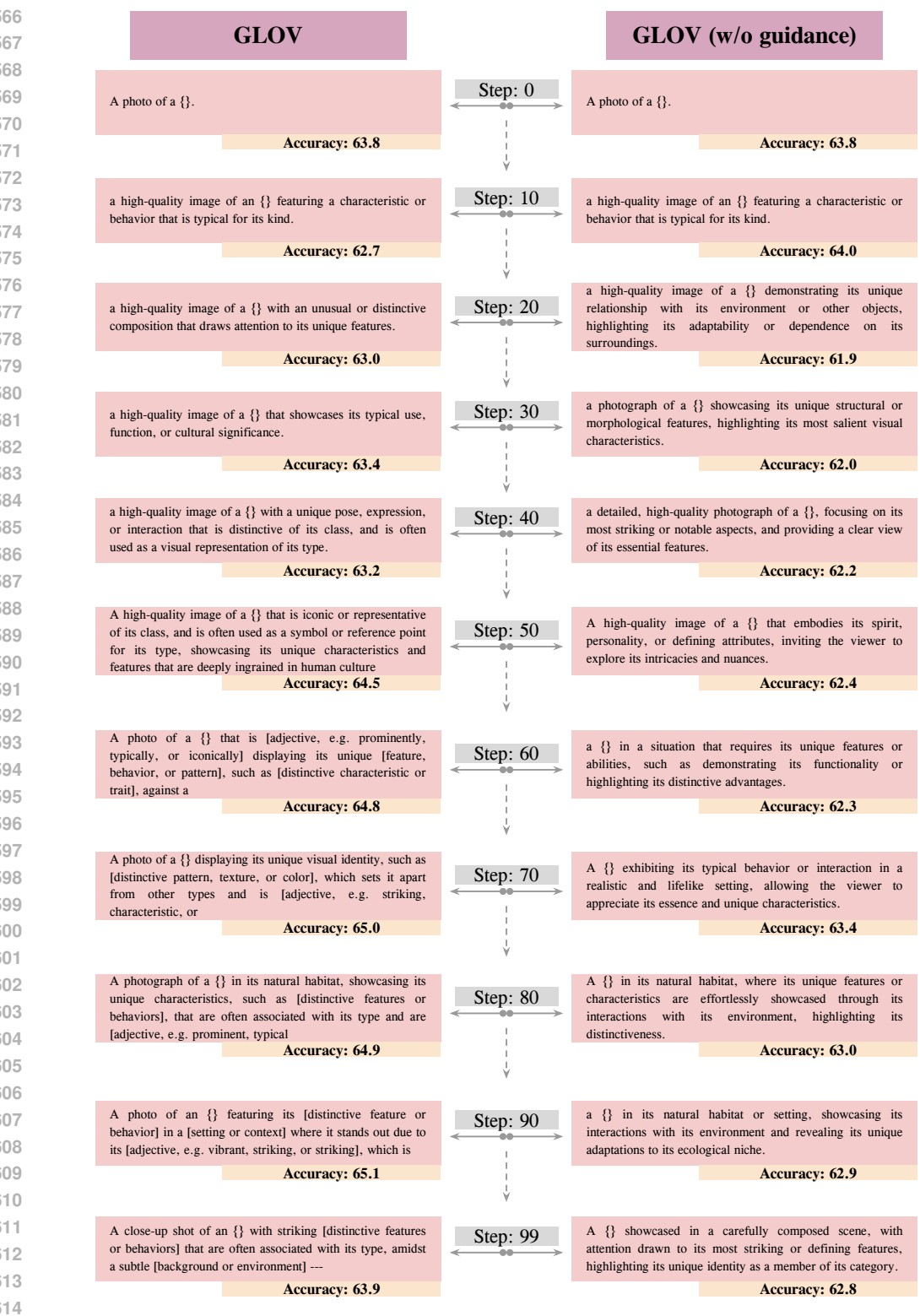

Figure 8: **Prompt evolution for CLIP**. We provide the highest performing prompt (on the 1-shot train set) discovered by our GLOV at different optimization steps for the ImageNet dataset.

