# OpenReview forum: "GLOV: Guided Large Language Models as Implicit Optimizers for Vision Language Models"
_ICLR.cc/2025/Conference — Submitted to ICLR 2025_

### Official Review · Reviewer_ZKNZ · 2024-10-29

**Soundness:** 2
**Presentation:** 2
**Contribution:** 1
**Rating:** 3
**Confidence:** 5

**Summary:**

This paper aims to improve the VLMs’ performance in downstream tasks. One prompt optimization method, namely GLOV, is proposed by meta-prompting LLM to choose the type of language structure preferred by the downstream VLM. At each optimization step, one embedding space steering methodology is used to bound the outputs more strictly. Empirical experiments with different VLM architectures on multiple datasets show the GLOV’s effectiveness.

**Strengths:**

[+] Improving the generalization of VLM in downstream tasks with low cost (parameters, fine-tuning data, training speed) is one practical topic that deserves further explorations.

[+] The paper is easy to follow and understand, having clear logic.

[+] Some experiments are conducted to demonstrate the idea’s performance, as well as the values of optimal prompt search.

**Weaknesses:**

[-] Novelty. Meta-prompts have been introduced by [1], while this paper expands the idea to a few-shot training data, which is rather trivial and brings minor technological contributions to the community. For the designs of meta-prompts, how to verify that this solution is optimal?

[-] Impracticality. As we all know, due to the autoregressive paradigm, the LLM inference requires a significant amount of cost compared to encoder-based single-forward models. Thus, employing LLMs in an iterative workflow, to find optimal language prompts seems unrealistic for real-world applications.

[-] Unknown efficacy. In the main manuscript, only the performance of downstream tasks is reported, without any computational/time complexity. The reviewer suggests to provide the inference time (in seconds) and the required GPU memory (in GB) for all methods in Table 1-2 to clarify its practical value.

[-] Incomplete comparisons. To improve the model performance on downstream tasks, one popular and effective idea is parameter efficient fine-tuning (PET), such as prompt-tuning, LoRA, and adapter, which has shown impressive few-shot capability. In Table 5 of the supplementary materials, CoOp performs even worse than CLIP, which is surprisingly and suspiciously. It is necessary to compare PET with the proposed method, in terms of performance, parameters, training time, and inference speed of under the same settings.


[1] Meta-Prompting for Automating Zero-shot Visual Recognition with LLMs.

**Questions:**

For few-shot learning, the impact of uniquely labeled data on performance is significant. In this paper, how to select this sample to ensure that the reported results are statistically significant rather than random? What is the variance of performance if five times of experiments are conducted?

---

> ### Author Response · Authors · 2024-11-20
> **Rebuttal (1/2)**
>
> Thank you for the time and effort spent in reviewing our paper. In the following, we reply to all the concerns raised in the review.
>
> **W1 - Novelty:** We would like to stress the fact that our GLOV only builds upon [1] but introduces many novel aspects as listed below:
>
> - *Iterative Refinement.* Only the conceptual idea for meta-prompting is inspired by [1]. We restructure it to fit the needs of our task of iterative optimization by providing $top_k$ and $bottom_k$ (in-context) prompts with their effectiveness for generating new (more tailored) solutions for the downstream task. The original meta-prompt [1] does not iteratively refine the prompts. Our GLOV is centered around iterative refinement. The results show that this iterative refinement helps to improve the results over [1] by 2.5% (compared with GLOV - w/o guidance), when averaged over the 16 datasets. These results highlight the effectiveness of our modifications to the original meta prompt and the iterative refinement scheme.
>
> - *Novel Guidance Scheme.* To further bias the output language structure strictly towards the task of interest we have proposed a novel embedding space guidance scheme. The results show that the embedding space guidance can improve over the vanilla GLOV and can further boost the performance w.r.t [1].
> For example, by applying our guidance scheme we outperform GLOV (w/o guidance) by 2.6% and [1] by 5.1% when the results are averaged over the 16 datasets. These results further signify the importance of the novel embedding space guidance scheme.
>
> - *Downstream Tasks.* The applicability of [1] is restricted to only the task of object classification since it was designed for that particular task. On the other hand, our GLOV has widespread utility. Along with evaluating the classical task of object recognition, we have also evaluated the VQA task in Table 3 of our submitted manuscript. Further, in this rebuttal, we also extend the downstream task to general Visual Question Answering and enhancing the safety of the models. The results for these tasks are provided as the response to the **W2** of reviewer $qqDY$.
>
> **W2 - Impracticality:** We would like to point out that finding effective prompts for downstream tasks is an active area of research [2, 3, 4, 5]. In our work, we take this burden off the humans to find the effective prompts for diverse downstream visual tasks and instead propose a generic optimization method that can be applied to find effective prompts for many downstream tasks, such as image classification, visual question answering, and enhancing the safety of models.
> We believe that the diverse nature of tasks that can be addressed by our method makes it extremely practical for real-world use cases.
>
> Furthermore, we would also like to point out that our method is designed to be applied offline -- where the optimal prompts can be found once and can be used without incurring extra overhead during test time.
> Even the training is very cheap since it only requires 1-shot training data for refining the prompts w.r.t a fitness function.
> We further request the reviewer to consider the newly evaluated (sub-shot) setting (that has fixed cost, independent of target dataset size) listed in response to the (next) W3, which further enhances the practical aspects of our approach.
>
> **W3 - Efficacy of the method and GPU memory:** Thank you for the comment. Here, we would like to point out that our GLOV only discovers optimal prompts during an offline training phase. Once the prompt is discovered, there is no added overhead during the evaluation (at test time).
>
> To further expand, for CLIP, at test time there is no added cost for GLOV other than ensembling of prompts -- which are *only* 3 -- discovered during the optimization procedure.
> Whereas, the original CLIP paper proposed, and the recommended way to use CLIP is, to ensemble $80$ prompts (for ImageNet), which is much more expensive.
> Our ensemble of *only* $3$ prompts also outperforms CLIP’s costly ensembling of $80$ prompts by $1.2\%$ for ImageNet and $1.7\%$ on average, over the 16 datasets.
> For LlaVa-type models, we only use a single best prompt because ensembling is not feasible for those models.
>
> We would also like to thank the reviewer for raising this concern because it indirectly led us to decrease the training time by many folds as well, with negligible loss in accuracy, and further helped us to discover a unique property that is specific to only GLOV, in contrast to other few-shot learning methods.
> Specifically, we found that we can only use a fraction of the total number of classes for optimizing the prompts (during the iterative optimization) with very minute degradation in performance.
> Essentially, making our GLOV a *sub-shot* method (that is, capable of operating at a fixed cost for datasets with many classes by using only a tiny fraction of the classes for optimization).
>
> --- continued in next comment ---

---

> ### Author Response · Authors · 2024-11-20
> **Rebuttal (2/2)**
>
> Below we report these results for LlaVA-OV on ImageNet and Imagenet-R:
>
> - ImageNet:
>
> | classes for optimization | 1000| 50 | 20| 15 |10|LlaVA-OV|
> |----------|----------|----------|----------|----------|----------|----------|
> | accuracy (%)   | 51.7   | 49.9   |47.0|46.6|44.8|36.5
>
> - ImageNet-R:
>
> | classes for optimization | 200| 25 | 20| 15 |10|5|LlaVA-OV|
> |----------|----------|----------|----------|----------|----------|----------|----------|
> | accuracy (%)   | 77.6 | 73.2 | 72.6 | 71.1 | 68.6 | 54.3 | 52.1     |
>
> These results show that our GLOV can show strong performance gains over the base model (LlaVA-OV) in this new *sub-shot* setting.
> For example, (for ImageNet) by only compromising 1.8% accuracy, our GLOV can find the optimal prompt (by using only 50 classes as compared to the full set of 1000) in less than 3 hours.
> Further, hinting that the discovered prompts can be generalized by having access to only a few classes. We believe that these experiments further enhance the practicability of our approach by reducing the time required for optimization by many folds.
>
> We also point out that in LN 364-365, we provide the GPU memory required to run our experiments.
> Specifically, our prompt search for CLIP can run on a single NVIDIA 3090 (24 GB) GPU and for LlaVA a single A40 (48 GB) suffices.
> The total number of GPUs used for our experiments is reported in the appendix (LN 820-822).
> During test time, there is no extra overhead, as previously mentioned.
> All our evaluations and of all other methods, fit on the same GPUs mentioned above.
>
> **W4 - Comparisons with PEFT:** In the main manuscript we report 1-shot results obtained with a parameter efficient fine-tuning (PEFT) based method -- CoOp [6].
> These results are consistent with the 1-shot results reported in the literature [7] for the same CLIP backbone (ViT B/32).
> We ran their (CoOp [6]) official code-base with the settings listed in the paper.
> One reason for the low results of CoOp can be severe over-fitting due to the extremely low data regime.
>
> As suggested by the reviewer, we also evaluate LORA by fine-tuning it on 1-shot data and find that when LORA is applied to all the matrices of the encoders, it results in severe overfitting and the results even degrade lower than the baseline model.
> We further evaluate by applying LORA fine-tuning by only applying it to the attention blocks and find that the results are better but still our method outperforms this fine-tuning variant as well.
>
> These results suggest that in extremely low data regimes, few-shot methods might not fare well due to overfitting. On the other hand, our GLOV is *parameter-update-free* and only relies on suitable prompts for performance gains, thus, making it more effective for low data.
>
> The results are listed as follows:
>
> |               | ImageNet | ImageNetA | ImageNetS | UCF101 |
> |---------------|----------|-----------|-----------|--------|
> | CLIP          | 61.9     | 28.2      | 40.3      | 60.4   |
> | CoOp          | 60.6     | 24.5      | 39.9      | 63.8   |
> | LORA (all)    | 59.9     | 27.1      | 37.1      | 58.2   |
> | LORA (attention) | 62.6  | 30.1      | 40.5      | 62.1   |
> | GLOV          | 64.5     | 32.5      | 43.0      | 63.8   |
>
>
> **Q - Selection of few-shot data and variance of results:**
> Consistent with the few-shot literature [6] we do not control the few-shot samples chosen and in our experiments sample the 1-shot training data randomly.
> As suggested by the reviewer, in the following table we report the mean and the variance over $5$ independent runs. The results indicate that the performance improvements are indeed statistically significant.
>
> |         | ImageNet    | ImageNetA   | ImageNetS    | UCF101    |
> |---------|-------|-------|--------|--------|
> | CLIP    | 61.9  | 28.2  | 40.3   | 60.4   |
> | GLOV    | 64.3 ± 0.43 | 32 ± 0.69 | 43.1 ± 0.25 | 63.9 ± 0.34 |
>
>
> [1] Meta-Prompting for Automating Zero-shot Visual Recognition with LLMs
>
> [2] Chain-of-Thought Prompting Elicits Reasoning in Large Language Models
>
> [3] Tree of Thoughts: Deliberate Problem Solving with Large Language Models
>
> [4] Maieutic Prompting: Logically Consistent Reasoning with Recursive Explanations
>
> [5] Complexity-Based Prompting for Multi-Step Reasoning
>
> [6] Learning to Prompt for Vision-Language Models
>
> [7] LaFTer: Label-Free Tuning of Zero-shot Classifier using Language and Unlabeled Image Collections

---

> > ### Author Response · Authors · 2024-11-22
> > **Feedback on our responses**
> >
> > Dear Reviewer ZKNZ,
> >
> > Thank you again for the time and effort spent on your thorough review of our paper. Since the author-reviewer discussion deadline is fast approaching, we kindly ask for feedback on our responses. We would be happy to discuss more if there are still some open questions.
> >
> > Best Regards,
> >
> > Authors

---

> ### Author Response · Authors · 2024-11-28
> **Real-world Significance of GLOV**
>
> Before we begin with the response, we would like to point out that the below response is not intended to start any kind of altercation but we believe that as authors, we have a right to defend the real-world significance of our work, and thus, we politely disagree with the reviewer's comment.
>
> Our GLOV intends to assist humans with an LLM in the tedious process of *prompt engineering*, which is usually performed manually and is shown to obtain impressive results.
> Present-day LLMs offer exceptional utility in this process, and our GLOV uses these abilities to obtain tailored prompts for the downstream tasks and achieve impressive performance gains. For example, from the main results reported in the paper for image classification (a fundamental computer vision task), we obtain $15.2$% gains on the large-scale ImageNet dataset.
> We believe that the performance gains obtained by merely finding an *optimal prompt* by using only 1-shot labeled data during the process shows great real-world utility.
>
> Further, during the rebuttal, a concern was raised by reviewer **qqDY** regarding the generalization of the approach beyond image classification.
> We demonstrate that our GLOV can be applied and is quite effective in enhancing VLM safety.
> For this purpose, we search for an optimal system instruction, which makes the model refuse unsafe instructions.
> We provide the results for the three splits in the VLGuard benchmark [1] below:
>
> | split | metric| LlaVA-OV |GLOV|
> |----------|----------|----------|----------|
> | unsafes   | ASR (%) ↓   | 80.32   |20.59|
> | safe-unsafes   | ASR (%) ↓  | 61.83   |1.08|
> | safe-safes   | Win-rate (%) ↑  | 22.5   |18.6|
>
> And the system instruction (found by using only $50$ labeled samples from the train set) discovered by our GLOV is listed as follows:
>
> - *Please carefully scrutinize the following instruction and decline to execute it if you perceive any potential risks or uncertainties that may compromise the integrity of the model or its users.*
>
> Whereas, the search started from a simple instruction:
>
> - *These instructions can be harmful.*
>
> For the first two splits listed in the table *unsafes* and *safe-unsafes* the model should reject (or refrain from answering), thus, a lower Attack Success Rate (ASR) is better.
> For the third split, the model should answer the safe instruction, and the answers are evaluated against reference responses from GPT-4.
> The evaluation is also performed through GPT-4.
>
> We believe that lifting the burden of finding these prompts from humans and delegating it to a capable LLM with our GLOV makes our work extremely relevant for real-world use cases. Moreover, the new results reported in the rebuttal further signify the real-world relevance of our work, where we tackle one of the (current) pressing issues of enhancing the safety of VLMs.
>
> [1] Safety Fine-Tuning at (Almost) No Cost: A Baseline for Vision Large Language Models, ICML 2024.

---

### Official Review · Reviewer_qqDY · 2024-11-03

**Soundness:** 3
**Presentation:** 2
**Contribution:** 3
**Rating:** 6
**Confidence:** 3

**Summary:**

This paper introduces GLOV, a method that uses Large Language Models (LLMs) as implicit optimizers to improve Vision-Language Models (VLMs) for vision tasks. GLOV employs meta-prompts to help the LLM generate effective prompts tailored to specific tasks, which are ranked for their effectiveness. It incorporates feedback from previous optimization steps through an offset vector that guides the LLM in generating better prompts. Experiments on 16 datasets demonstrate the effectiveness of proposed methods.

**Strengths:**

- This paper introduces an interesting framework, GLOV, to enhance VLMs for image classification.
- The use of meta-prompts and LLM steering provides fresh insights in this field.
- Experimental results demonstrate the effectiveness of the proposed methods compared to  baselines.

**Weaknesses:**

- Lack of comparison. While GLOV approaches image classification from a novel perspective, previous works [1,2,3] in this area already achieved promising results with lower costs. Could authors provide a comparison of GLOV with these baselines?
- The generalization ability of GLOV is not clear. The authors demonstrated the effectiveness of the proposed methods on VLM image classification and visual question answers under the same topic. However, if the GLOV is not competitive compared with other works focused on VLM image classification[1,2,3]. Could authors prove the generalization ability of GLOV on visual tasks beyond image classification?
- Clarity of Figure 2: The method overview in Figure 2 is difficult to understand. If the authors could clearly show the flow of iterative optimization, the methods would be easier to follow.
- Lack of inference cost comparison: Could the authors show the curve of iteration steps and inference time to illustrate the trade-off between performance and cost in GLOV?


Reference:
[1]AWT: Transferring Vision-Language Models via Augmentation, Weighting, and Transportation
[2]Sus-x: Training-free name-only transfer of vision-language models
[3]Tip-Adapter: Training-free CLIP-Adapter for Better Vision-Language Modeling

**Questions:**

Please kindly answer the question in the weakness section.

---

> ### Author Response · Authors · 2024-11-20
> **Rebuttal (1/2)**
>
> Thank you for the time and effort spent in reviewing our paper. In the following, we reply to all the concerns raised in the review.
>
> **W1 - Lack of comparisons:**
> Thank you for the suggestion. In the following, we provide comparisons with three methods on the ImageNet dataset while using the CLIP ViT-B/32 backbone.
>
> | method: | CLIP| TIP [3] | Sus-X [2] | AWT [1] |GLOV|
> |----------|----------|----------|----------|----------|----------|
> | results   | 61.9   | 62.32   |64.73|64.89|64.54
>
> As we can see GLOV outperforms TIP, and performs on par with AWT and Sus-X. However, there are important distinctions between AWT, Sus-X and GLOV in terms of the task definition (operating assumptions) that make them (AWT and Sus-X) not directly comparable to GLOV. In particular, these methods synthesize category-specific prompts (as opposed to GLOV which optimizes for shared prompts for all categories of the same task). Category-specific prompts are naturally more expensive both in terms of optimization as well as inference time. Additionally, as we discuss more extensively in a new experiment performed due to the reviewer's suggestion below, GLOV can successfully operate (almost without losing performance) even under significantly more challenging (and more practical) conditions of knowledge of only a small fraction of the downstream task classes.
> This is impossible for the other (TIP, Sus-X, AWT) methods that always require the a-priori knowledge of all categories for their operation. Moreover, Sus-X also strongly relies on an external diffusion model making it unfair to compare to GLOV.
> Finally, TIP, Sus-X, and AWT are only designed to work with encoder-only models (like CLIP) and cannot be directly applied to encoder-decoder models (like LlaVA-OV) as opposed to GLOV.
>
> More Specifically:
> - Sus-X and AWT require per-category (multiple) descriptions that are generated from an LLM (i.e., GPT).
> For large-scale datasets like ImageNet, containing 1000 categories, this can become prohibitively expensive.
> In contrast, our method does not require generating category-level descriptions but only finds optimal prompts (i.e., 3) that are generalizable for the entire dataset.
>
> - Sus-X also distills knowledge from an external model – requiring generating a support set (of images for all categories in the dataset) through a stable diffusion model, which is an external model and further enhances the cost of using their method.
>
> - All these methods are only suitable for object classification through dual-encoder models (e.g., CLIP) because they all require an output probability distribution over the entire class space to work, which cannot be obtained by the generative (encoder-decoder models like LlaVA). In contrast, our method extends beyond dual-encoder models and is also applicable to other open-ended visual question-answering tasks (in addition to image classification reported in the submitted manuscript), as well as also applicable for enhancing safety of encoder-decoder models.
> These results are discussed in the response to the *weakness 2* below.
>
>
> **W2 - Generalization of GLOV to other tasks:** In the submitted manuscript, we evaluated our GLOV for the task of VQA in the context of image classification on the FOCI benchmark.
> Here, in response to the reviewer's comment, we further extend the evaluation of GLOV to $3$ additional tasks.
>
> The first two tasks are: ChartQA (VQA for charts) and GQA (compositional questions created from image scene graphs).
> Our GLOV can provide an improvement of 1.0% (on average) for these tasks, where it improves the LlaVA model from 79.44% --> 80.21 for ChartQA and 61.13% --> 62.21% for the GQA dataset. These results highlight the applicability of GLOV to general VQA tasks, different from the image classification.
> For both these datasets, GLOV is tasked with finding an instruction, which is prepended to the actual question.
> These prompts are added to the appendix of the updated manuscript.
>
> Furthermore, we demonstrate that our GLOV can be applied and is quite effective in enhancing VLM safety. For this purpose, we search for an optimal system instruction, which makes the model refuse unsafe instructions.
> We provide the results for the three splits in the VLGuard benchmark [4] below:
>
> | split | metric| LlaVA-OV |GLOV|
> |----------|----------|----------|----------|
> | unsafes   | ASR (%) ↓   | 80.32   |20.59|
> | safe-unsafes   | ASR (%) ↓  | 61.83   |1.08|
> | safe-safes   | Win-rate (%) ↑  | 22.5   |18.6|
>
> ---- continued in next comment ----

---

> > ### Author Response · Authors · 2024-11-20
> > **Rebuttal (2/2)**
> >
> > And the system instruction discovered by our GLOV is listed as follows:
> >
> > - *Please carefully scrutinize the following instruction and decline to execute it if you perceive any potential risks or uncertainties that may compromise the integrity of the model or its users.*
> >
> > Whereas, the search started from a simple instruction:
> >
> > - *These instructions can be harmful.*
> >
> > For the first two splits listed in the table *unsafes* and *safe-unsafes* the model should reject (or refrain from answering), thus, a lower Attack Success Rate (ASR) is better.
> > For the third split, the model should answer the safe instruction, and the answers are evaluated against reference responses from GPT-4.
> > The evaluation is also performed through GPT-4.
> >
> > From the results we observe that our GLOV can reduce the ASR for the *unsafe* instructions by ~60%, whereas for the *safe-unsafe* split, the safety prompt discovered by GLOV can bring down the ASR to an extremely low value of 1.08% -- showing that our system prompt can induce the model with abilities to critically analyze the *harmful* instructions and reject (almost) all of them.
> > We also see that the ASR is brought down on the cost of minimum loss in the win rate on the *safe-safes* subset.
> >
> > We would again like to point out that [1, 2, 3] are not suitable for these tasks, whereas the applicability of GLOV is general and extends much beyond *only* the dual encoder models and image classification.
> >
> >
> >
> > **W3 - Clarity of figure 2:** Thank you for the valuable suggestion. We have explicitly added the legends to the main Figure 2, to show the *main optimization loop* and the *helper inputs*. Please let us know if these changes make it clear, or we should further iterate over the figure to make it more clear.
> >
> > **W4 - Inference and cost comparison:** We would like to point out that our GLOV *only* finds optimal prompts on a 1-shot train set and does not iteratively find prompts during the test phase.
> > Thus, at inference, there is no added cost other than ensembling of prompts (for CLIP) -- which are *only* 3.
> > Whereas the original CLIP paper proposed to ensemble $80$ prompts, which is much more expensive.
> > Our ensemble of $3$ prompts also outperforms CLIP’s more costly ensembling of 80 prompts.
> > For LlaVA-type models, we only use a single best prompt because ensembling is commonly not used for those models.
> >
> > Thanks to the reviewer's suggestion, we also had a chance to analyze and reduce the cost during the (training) optimization for GLOV by many folds and found that we can only use a fraction of the total number of classes for optimizing the prompts with very minute degradation in performance.
> > Essentially, making our GLOV a *sub-shot* method capable of strong generalization from observing only a small fraction of dataset classes.
> > Below we report these results for LlaVA-OV on ImageNet:
> >
> > | classes for optimization | 1000| 50 | 20| 15 |10|LlaVA-OV|
> > |----------|----------|----------|----------|----------|----------|----------|
> > | accuracy (%)   | 51.7   | 49.9   |47.0|46.6|44.8|36.5
> >
> > These results show that our GLOV can show strong performance gains in this new *sub-shot* setting.
> > For example, by only compromising 1.8% accuracy, our GLOV can find the optimal prompt (by using only 50 classes as compared to 1000) in less than 3 hours.
> > Further, hinting that the found prompts can work by having access to only a few classes.
> > This shows the strong generalization ability of GLOV, showing that it can be used at a fixed cost regardless of the size of the targeted dataset, a property that is unique to GLOV, in contrast to [1, 2, 3] that requires knowledge of all classes for their optimization.
> >
> >
> > [1] AWT: Transferring Vision-Language Models via Augmentation, Weighting, and Transportation
> >
> > [2] Sus-x: Training-free name-only transfer of vision-language models
> >
> > [3] Tip-Adapter: Training-free CLIP-Adapter for Better Vision-Language Modeling
> >
> > [4] Safety Fine-Tuning at (Almost) No Cost: A Baseline for Vision Large Language Models.

---

> > > ### Author Response · Authors · 2024-11-22
> > > **Feedback on our responses**
> > >
> > > Dear Reviewer qqDY,
> > >
> > > Thank you again for the time and effort spent on your thorough review of our paper. Since the author-reviewer discussion deadline is fast approaching, we kindly ask for feedback on our responses. We would be happy to discuss more if there are still some open questions.
> > >
> > > Best Regards,
> > >
> > > Authors

---

> > > > ### Comment · Reviewer_qqDY · 2024-11-23
> > > >
> > > > Thanks for the authors' detailed response. I appreciate the effort you have put into addressing the concerns raised. However, I still have a few follow-up questions.
> > > > - **W1 Lack of comparisons**: Could authors provide more details about the comparison results, such as the number of shots for the baselines and where is the source of these numbers?
> > > > - **W2 - Generalization of GLOV to other tasks**: I noticed that the prompts for ChartQA and GQA are specifically tailored, as detailed in the updated appendix. This reinforces my concerns regarding the generalization of the proposed method. Could the authors provide results on at least one general MLLM benchmark, such as MME, SeedBench, or MMStar? If time does not permit additional experiments, could the authors provide a justification regarding the generalization issue, particularly addressing whether the proposed method has specific requirements or constraints for downstream tasks?

---

> ### Author Response · Authors · 2024-11-23
> **Response to additional questions**
>
> Dear Reviewer qqDY,
>
> Thank you for the acknowledgment of our rebuttal. Here we provide the answer and discussion to the two questions:
>
> **Q1.** For TIP and AWT - we used 1-shot data from the train set of ImageNet, which is consistent with the setting we follow in our work. We use the ViT-B/32 architecture for the main results in our paper (Table 1). TIP tests ViT-B/32 architecture in their paper but does not provide individual results for the datasets, whereas, AWT does not test with ViT-B/32 architecture. To obtain the results with the same backbone we run their official codebase, using the setting mentioned in their paper.
> For few-shot results with AWT, we used the optimal multi-modal adapter as listed in their appendix (B3), while for TIP we followed all settings from their code base. On the other hand, for SuS-X, the authors provide the results with different backbones in Table 20 (page 27) of the appendix of their Arxiv submission (https://arxiv.org/pdf/2211.16198) and we report the ViT-B/32 results from there.
>
> **Q2.** During the rebuttal, we have provided results for $3$ additional tasks: ChartQA, GQA, and (MMLM Safety) VLGuard. The results show that the proposed method can generalize beyond image classification to other VQA tasks and also a very relevant task of making the VLM’s responses safe.
>
> While we acknowledge the observation of the reviewer regarding ‘tailored’ prompts for ChartQA and GQA - we would like to point out that the motivation of our GLOV is to optimize for task-specific prompts to be applied to instances of a particular task. On the other hand, general benchmarks like SEED and MME might require optimization for instance specific prompts, which is left as an exciting future work direction not touched upon in our current work.
>
> We would further like to emphasize, that the optimization of task-specific prompts can be extremely helpful in achieving specific goals, e.g., improving VLM safety (as we show in the rebuttal).
>
> We will add these discussion points to the updated manuscript. Furthermore, we would also be happy to discuss if there are more open points from your end. Thank you again for your time!

---

> > ### Comment · Reviewer_qqDY · 2024-11-26
> >
> > Thanks for the authors' responses. My concern is addressed for the **W1**, and I recommend the authors add comparison results with recent baselines in the revised version.
> > For **W2**, the proposed method is restricted to simple and task-specific scenarios such as image classification and the generalization to complex scenarios such as VQA that require instruction following ability seems rather limited.
> > However, I think the contribution of this work is enough and I will raise my score.

---

> > > ### Author Response · Authors · 2024-11-26
> > > **Thank you!**
> > >
> > > We sincerely thank you for the time and effort spent during the review period. We will add the new comparisons to the updated manuscript!
> > >
> > > Best,
> > > Authors.

---

### Official Review · Reviewer_CSHE · 2024-11-04

**Soundness:** 3
**Presentation:** 3
**Contribution:** 3
**Rating:** 8
**Confidence:** 3

**Summary:**

This paper proposes a novel framework GLOV, which enables LLMs to act as implicit optimizers for VLMs to enhance downstream vision tasks. Experiments highlight the effectiveness of GLOV for both dual-encoder and encoder-decoder architectures.

**Strengths:**

* The introduction of steering LLM response during the prompt optimization process presents a novel and effective methodology.
* The steering strategy designed by analogy with the gradient update process, while lacking a rigorous theoretical basis, conforms well to engineering intuition.
* The article is highly readable, featuring a well-defined and clear structure.

**Weaknesses:**

* The applicability of GLOV optimization to a given task is constrained by the existence of an objective fitness function for the task.
* For encoder-decoder models such as LLaVA, it seems the VLM response has to be relatively concise in form. When dealing with complex responses (such as responses for image captioning tasks), the reliability of the sentence embeddings computed via Equation 3 remains unverified.

**Questions:**

* In the context of encoder-decoder architectures, is there a potential for the emergence of lengthy and ambiguous symbolic representations during the optimization process? Furthermore, what measures can be implemented to ensure the efficacy of sentence transformers under such circumstances?

* The reviewer expresses concern that the adoption of top-k and bottom-k approaches for in-context examples may result in a significant disparity between positive and negative samples in the later stages of training, potentially hindering the model to learn subtle prompt refinements akin to the challenges posed by consistently employing a large learning rate in gradient-based optimization. Consequently, the reviewer prefers implementing a dynamic selection threshold as a more reasonable choice. Any insights regarding the current strategy would enhance the understanding of the paper.

* Similarly, in the steering of LLM, would it be more judicious to dynamically modify the rank interval between the positive (P+) and negative (P-)?

---

> ### Author Response · Authors · 2024-11-20
> **Rebuttal**
>
> Thank you for the time and effort spent in reviewing our paper. In the following, we reply to all the concerns raised in the review.
>
> **W1 - GLOV optimization is constrained by an objective function:** We acknowledge the comment by the reviewer. However, we would like to point out that our motivation is parallel to the traditional gradient-based optimization and we also share the same philosophy as [1], which proposes to optimize prompts for natural language tasks.
> For any kind of optimization, as the reviewer might also acknowledge, there is a need for an objective function to analyze the *goodness* of the learning process.
> Our work comes with a similar constraint.
>
> **W2 - Reliability of sentence embeddings (from Eq. 3) for complex tasks:**
> We would like to point out that in Eq. 3 - the sentence embeddings are not computed from the outputs of the VLM, but rather the outputs from the LLMs, which are natural language prompts, used for the *guidance* in the optimization procedure.
>
> To analyze the reliability of the sentence embeddings, we evaluated the output responses from different layers in the LLM by linear probing, for the task of SST-5 (sentiment classification) and reported these results in Figure 4 (left) of the main manuscript.
> We find that the best accuracy obtained for this task by evaluating the sentence embeddings from the middle layers is 56.7%.
> This accuracy is close to the state-of-the-art results obtained by [2] (59.8%) -- by a dedicated sentence embedding model.
> These results indicate that the sentence embeddings obtained through our method are semantically meaningful and reliable.
>
>
> **Q1 - Symbolic representations for encoder-decoder models:** We agree with the reviewer that sometimes the symbolic representation can also be lengthy.
> For this work, we chose a state-of-the-art open-source sentence embedding model from HuggingFace [3] which has been widely used by the community and is known to be reliable for extracting meaningful sentence embeddings even from long text.
> Furthermore, this type of evaluation (for encoder-decoder models) has also been extensively used in prior works [4, 5, 6], that also use an embedding model.
> The strong capabilities of these models can help them to extract meaningful semantics even from long text.
>
> **Q2 - Choice of top-k and bottom-k:** The motivation behind the current choice of the $top_k$ and $bottom_k$ in-context examples is that we intend to provide contrasting examples to the LLM from the opposite end of the spectrum (of *goodness* and *badness*) so that the LLM can make sense of what are the type of responses preferred by the downstream VLM.
> We have added this motivation in the method section of the updated manuscript.
>
> We also want to thank the reviewer for suggesting dynamic thresholding for the choice of $top_k$ and $bottom_k$, which can essentially make the learning algorithm more robust if we mix it with the current strategy.
> However, currently, we leave it as an exciting future work direction.
>
> **Q3 - Dynamic modification of rank interval for P+ and P-:** Thank you again for the suggestion and this can definitely be one of the future directions to improve the current algorithm.
>
> For the current optimization algorithm: In our initial experiments we tried different ways to choose $P_+$ and $P_-$, such as selecting the best and worst prompts as *negative* and *positive*, however, that resulted in unstable optimization.
> That could be because of the larger values in the guidance vector.
> This is the reason, we chose to select points closer to each other (best and second best).
> The motivation (also present in LN 299-304) is that we compute a form of a gradient-like differential between averages of token hidden states, intuitively trying to identify a characteristic of task-specific improvement.
> Thus, the intuition behind computing the differential between the best and the second best (in terms of fitness) is to make it between points closest to the maximal value of the objective -- which is a common mathematical intuition.
>
>
> [1] Large Language Models as Optimizers
>
> [2] An Algorithm for Routing Vectors in Sequences
>
> [3] https://huggingface.co/sentence-transformers/all-mpnet-base-v2
>
> [4] Vocabulary-free Image Classification
>
> [5] Democratizing Fine-grained Visual Recognition with Large Language Models
>
> [6] RAR: Retrieving And Ranking Augmented MLLMs for Visual Recognition

---

### Official Review · Reviewer_h9hv · 2024-11-08

**Soundness:** 3
**Presentation:** 4
**Contribution:** 3
**Rating:** 6
**Confidence:** 4

**Summary:**

The paper proposes GLOV, an LLM-assisted framework for automatic optimization of VLM prompts for specific downstream tasks. Specifically, an LLM is meta-prompted to generate downstream VLM prompts based on task description and in-context examples from previously generated prompts as feedbacks. On top of the meta prompt, the paper also applies feature-level guidance, i.e., the difference of sentence embedding from bad prompts to good prompts, as a second measure to push the LLM output towards the more effective direction. The proposed method is evaluated mainly on 16 few-shot classification tasks and shows improvement over baselines, while preliminary results on VQA are also provided.

**Strengths:**

* The motivation is sound and clear.

* The experimental results are transparent, with search outcomes in the appendix and code release promised.

* The prompts generated by search are shown to generalize within the same category (dual encoder) of VLMs.

**Weaknesses:**

* **Feature-level guidance poses white-box LLM constraint**: Despite the feature guidance being novel for VLM prompt optimization, it requires access to LLM intermediate features, which could be hard to obtain given that many strong LLMs are either closed-source or too large to easily run locally. This could be a hedge against the advantages of black-box methods, as the LLM intermediate features could be even harder to get than parameters or gradients of VLMs in many cases.

* **Sensitivity to LLM choices is not clear**: While the proposed method shows clear improvements, it would make the argument stronger if more evidence could be given showing that the reasoning of the LLM indeed plays an important role, especially with the fluctuation (e.g., Fig 1, 3, 6) of the results and the general impression that LLMs at 7B-level are not very good at reasoning or agent-like applications. One way to show this is higher accuracy or less steps to convergence with a stronger LLM.

**Questions:**

* **Clarity of Algorithm 1**: At lines 9, 12, 28, 29, it's unclear what is the meaning of the square brackets, given that $K$ is an integer according to the input. It's also not clear how the top-3 prompts used for ensemble are selected: Are they from the last step, a single best step, or all steps through out the search process?

* **Sensitivity to the hyper-parameters**: The LLM steering part introduces two hyper-parameters, layer $l$ and scaling factor $\alpha$. Are these hyper-parameters searched on one dataset and kept the same for the others, or searched independently on each dataset? How different could the optimal choices be over different datasets?

* **Additional placeholders in prompts**: Some searched prompts (e.g., at Ln 1187-1192) seem to contain additional placeholders (in angle brackets). Are they from additional metadata of the dataset? Is the search process altered in any way to account for these additional information?

* **Average accuracy in main tables** (e.g., Table 1, 2) would make it easier to cross-reference the results with the ablation studies (e.g., Table 4)

---

> ### Author Response · Authors · 2024-11-20
> **Rebuttal (1/2)**
>
> Thank you for the time and effort spent in reviewing our paper. In the following, we reply to all the concerns raised in the review.
>
> **W1  - Feature level guidance poses white-box LLM constraint:** We acknowledge the concern of the reviewer and indeed the guidance scheme can only be applied when the activations of the LLM are accessible. However, we would like to point out that in our paper we proposed two variants of our approach, i.e., *with and without guidance*. Our experiments show that our white-box approach also fares well compared to baselines.
> For example, our GLOV (without guidance) achieves 2.5% and 19.0% average improvements (over 16 datasets) for the base CLIP and LLaVA One Vision models.
> Furthermore, our GLOV (without guidance) also outperforms the other white-box approach LLM-OPT by 1.2% (averaged over 16 datasets) for the CLIP models.
> These results highlight that our prompt optimization method can indeed be helpful in obtaining strong performance gains even when the activations of the LLM are not accessible.
>
> **W2 - Sensitivity to LLM choice is not clear:** Thank you for pointing it out. In our paper, we employed the Llama-3.0 (7B) model. At the time of performing the experiments, Llama-3.0 was one of the state-of-the-art open LLMs.
> Further, we chose the smallest variant (7B) to keep the cost of the optimization low.
> To respond to the reviewer's concern, we further evaluated our GLOV with the Llama-3.1 (70B) variant, which is considered one of the more capable LLMs in terms of instruction-following abilities.
> We find that the accuracy improves by employing a stronger (and larger) LLM.
> Below, we list the comparison of our GLOV with Llama-3.0 (7B) and Llama-3.1 (70B) for the CLIP ViT-B/32 model.
> For ensuring reproducibility, the prompts produced by Llama-3.1 (70B) are added to the appendix for inspection, in the updated version of our manuscript.
>
>
>
> | models | DescribableTextures | EuroSAT | ImageNetR |ImageNetA|UCF-101|
> |----------|----------|----------|----------|----------|----------|
> | CLIP    | 40.2   | 35.8   | 65.4   | 28.2 | 60.4|
> | GLOV (Llama-3-7b)    | 42.6   | 50.8   | 68.5   | 32.5|63.8|
> | GLOV (Llama-3.1-70b)    | 44.5   | 54.0   | 69.7   | 33.3|64.7|
>
> We would also like to thank the reviewer for this suggestion since it helps showing that our method scales well with LLM size increase.
>
> **Q1 - Clarity of Algorithm 1 and Choice of Prompts:** Thank you for pointing this out. We made a minor mistake when listing the algorithm in the submitted manuscript. $K$ is indeed an integer, representing the number of prompts generated at each iteration.
> In lines 9 and 12 - the list of prompts should be referenced to find the $positive$ and $negative$ prompts. Similarly, in lines 28 and 31, $K$ should be replaced with the list containing the $NewPrompts$ generated at a certain iteration. We have corrected this minor mistake in the updated algorithm in the appendix.
>
> We select the best prompts found (w.r.t the 1-shot train set accuracy) at a single iteration (step) during the optimization and not through all steps.
> We also list this in the LN 360-361 in the main manuscript, where we list the implementation details.
>
> **Q2 - Sensitivity to hyper-parameters:** For choosing the optimal layer $l$ for guidance we ran a sweep over different layers in the Llama model, *only* for the ImageNet dataset, and found that guidance on layer $17$ performs the best for the downstream recognition task.
> We also evaluated the sentence embeddings for the sentiment classification task (SST-5) through linear probing of different layers and again found that the middle layers in the Llama model performed best.
> These results were included in the original submitted manuscript: plotted in Figure 4 and discussed in the ablation section.
>
> To clarify, $alpha$ is not actually a hyperparameter but a parameter optimized by GLOV automatically on the (1-shot) respective training set for each task. The optimization is done by an alpha sweep on the 1-shot training set used for GLOV optimization.
> Thanks to the reviewer’s suggestion, we have done some analysis on the $alpha$ automatically found by GLOV. For encoder-only models, GLOV found: $alpha = 1.0$ to be beneficial for the fine-grained classification datasets (e.g., Stanford Cars, Oxford Flowers); $alpha = 0.75$ to benefit datasets consisting of natural images (e.g., ImageNet); and  $alpha = 0.25$ to be optimal for the out-of-distribution ImageNet variants like ImageNetA. For the encoder-decoder models (i.e., LlaVA-OV), GLOV found $alpha=1$ to be optimal for all datasets.

---

> > ### Author Response · Authors · 2024-11-20
> > **Rebuttal (2/2)**
> >
> > **Q3 - Additional placeholders in prompts:** We do not alter the search process in any way for specific datasets.
> > One strong point of our method is that it naturally finds the type of language preferred by the downstream VLM (in the mentioned case -- CLIP) and this language can also sometimes contain placeholders.
> > To further address the reviewer’s comment and analyze the effect of these placeholders on the performance, we manually removed the *angle brackets* in the prompts found for the Aircraft dataset (Ln 1170-1177 of the updated manuscript) and performed the classification.
> > We observe that the accuracy decreases by 1.1% (20.1% --> 19.0%) if we manually remove these placeholders.
> > These results further signify the effectiveness of our method, where the discovered prompts might not seem natural to human understanding but are preferred by the model. We will add this finding and discussion to the paper, thanks for suggesting it!
> >
> >
> > **Q4 - Average accuracy in main tables:** Thank you for pointing this out. We have added the mean accuracies for all baselines and the methods of comparison in the updated manuscript. We observe that our GLOV provides an improvement of 3.8% and 21.6% over CLIP and the LlaVA-OV model -- when the results are averaged over all the $16$ datasets.
> > Similarly, we obtain an average improvement of 1.7% and 5.1% for the two models, when compared with the strongest baseline.

---

> ### Author Response · Authors · 2024-11-22
> **Feedback on our responses**
>
> Dear Reviewer h9hv,
>
> Thank you again for the time and effort spent on your thorough review of our paper. Since the author-reviewer discussion deadline is fast approaching, we kindly ask for feedback on our responses. We would be happy to discuss more if there are still some open questions.
>
> Best Regards,
>
> Authors

---

> ### Comment · Reviewer_h9hv · 2024-11-25
>
> Thanks for the responses from the authors as well as the other reviewers. The authors' comments resolved most of my concerns. I find the results on Llama-3.1-70b especially inspiring as it demonstrates the scalability of the proposed method. I would encourage that results on all datasets with Llama-3.1-70b be included in the final draft to further show the robustness of the performance improvement, if time permits.
>
> Although I agree with other reviewers on some practical limitations of the work (e.g., time consuming), I would still acknowledge some value of this work given that the relevant research on LLM for optimization are at a fairly preliminary state.
>
> With these considerations, I will raised my score to be positive.

---

> > ### Author Response · Authors · 2024-11-25
> > **Thank you!**
> >
> > We sincerely thank you for the time and effort put in during the review period. We will add the new results to the updated manuscript!
> >
> > Best,
> > Authors.

---

### Author Response · Authors · 2024-11-20
**Global Response**

We sincerely thank the reviewers for the time and effort spent reviewing our paper and for their thoughtful feedback. The positive reception of our paper’s **novel methodology** (CSHE), particularly the use of meta-prompts and the **steering of LLM responses** (qqDY) during prompt optimization, is highly encouraging. Further, we are pleased that the **practical value** (ZKNZ) of improving VLM generalization with minimal resource costs was appreciated, along with the paper's **clarity and structure** (ZKNZ, CSHE). Additionally, the acknowledgment of **transparency of experiments** (h9hv) is also heartening.

Below, we respond to all the reviewers’ comments individually and hope to indulge in a constructive discussion during the author-reviewer discussion period.

We thank you again.

---

### Comment · Area_Chair_JPjw · 2024-12-03
**Discussion due soon**

Dear all reviewers,

Our reviewer-author discussion will end soon. For each of you, please check all the files and see if anything you'd like to discuss with authors.

Best,
Your AC

---

### Meta-Review · Area_Chair_JPjw · 2024-12-17

**Metareview:**

This paper proposes a GLOV method to enable LLM as an implicit optimizer for VLMs for downstream task improvement. It received mixed reviews initially. The reviewers raised technical unclear presentation, limited applications, lack of sufficient comparison, and limited novelty. During the rebuttal phase, the authors try to address these issues by providing more explanations and experimental results, which turn reviewers into positive. However, one reviewer [ZKNZ] points out the work is similar to an existing one [1] and the implacability remains.  Although authors try to further explain the novelty and real applications, the justification does not motivate [ZKNZ] to change the opinion. Overall, the AC has checked all the files, and find [ZKNZ] is reasonable for the novelty issue. There is not much significant novel design upon [1] as listed by [ZKNZ]. The iterative manner is prevalent in many pipelines and the guidance seems as a loss supervision, which is not something new. The AC agrees that authors bring GLOV to the VLM field, but the novelty still remains limited according to the current form. The authors shall further improve the current work and welcome to submit for the next venue.

**Additional Comments On Reviewer Discussion:**

[ZKNZ] mentions the novelty issue, real-world application issues. The authors respond by showing iterative refinement, and novel guidance scheme. These designs are commonly found in many studies. Also, the real practical usage depends on the VLMs (i.e., the sensitivity of different VLMs is indeed different upon the prompt with LLM guidance). The wide application usage is not sufficient demonstrated.

---

### Decision · Program_Chairs · 2025-01-22

Reject